



# The radiative impact of biomass burning aerosols on dust emissions over Namibia and the long-range transport of smoke observed during AEROCLO-sA

Cyrille Flamant[1], Jean-Pierre Chaboureau[2], Marco Gaetani[3], Kerstin Schepanski[4], and Paola Formenti[5]

[1]LATMOS, CNRS, Sorbonne Université, UVSQ, Paris, France
[2]LAERO, Université de Toulouse, CNRS, UT3, IRD, Toulouse, France
[3]Scuola Universitaria Superiore IUSS, Pavia, Italy
[4]Institute of Meteorology, Freie Universität Berlin, Berlin, Germany
[5]Université Paris Cité and Université Paris Est Créteil, CNRS, LISA, Paris, France

**Correspondence:** Cyrille Flamant (cyrille.flamant@latmos.ipsl.fr)

**Abstract.**

The radiative effects of biomass burning aerosols (BBAs) on low-level atmospheric circulation over southern Africa are investigated on 5 September 2017 during the Aerosols, Radiation and Clouds in southern Africa (AEROCLO-sA) field campaign. This is conducted using a variety of in situ and remote sensing observations, as well as 5-day twin ensemble simulations made with the Meso-NH mesoscale model, one including the direct and semi-direct radiative effects of aerosols and one in which these effects are not included. We show that the radiative impact of BBA building up over a period of 5 days in the Meso-NH simulations can lead to significantly different circulations at low- and mid-levels, thereby affecting dust emissions over southern Namibia and northwestern South Africa as well as the transport of BBA in a so-called "river of smoke". While most of the regional scale dynamics, thermodynamics and composition features are convincingly represented in the simulation with BBA radiative effects, neglecting the radiative impact of BBA leads to unrealistic representations of (i) the low-level jet (LLJ) over the plateau plateau, which is the main low-level dynamic feature fostering dust emission, and (ii) the mid-level dynamics pertaining to the transport of BBA from the fire-prone regions in the Tropics to the mid-latitudes. For instance, when the BBA radiative impacts are not included, the LLJ is too weak and not well established over night, and the developing convective planetary boundary layer (PBL) is too deep compared to observations. The deeper convective PBL over Etosha and surrounding areas is related to the enhanced anomalous upward motion caused by the eastern displacement of the river of smoke. This eastern displacement is, in turn, related to the weaker southerly African Easterly Jet. Both ensemble simulations provide clear evidence that the enhanced near surface extinction coefficient values detected from observations over Etosha are related to the downward mixing of BBA in the developing convective boundary layer rather that dust being emitted as a result of the LLJ breakdown after sunrise. This study suggests that the radiative effect of BBAs needs to be taken into account to properly forecast dust emissions in Namibia.





## 1 Introduction

In the austral dry season (Aug-Sep-Oct), southern Africa atmospheric composition at the regional scale is dominated by biomass burning aerosols (BBAs) and terrigenous aerosols (so-called mineral dust). While human controlled and accidental
forest fires are widespread across Angola, Zambia, Zimbabwe, Mozambique, the Democratic Republic of Congo, and South Africa (van der Werf et al., 2017a), dust sources are known to be concentrated in a few hotspots (Vickery et al., 2013): the Etosha Basin (Namibia), the Makgadikgadi Basin (Botswana), the Namib Coastal Sources (Namibia), and the South Western Kalahari Sources (across northwestern South Africa and southeastern Namibia). Peak emissions from these hot spots are generally observed in August and September. It is worth noting that all of the fire-prone regions and most of the dust hot spots
are located on the high southern Africa plateau, which has a mean elevation of about 1 km above mean sea level (amsl). The plateau is bounded west, south and east by bands of high ground which fall steeply to the coasts. The Great Escarpment is a major topographical feature that consists of steep slopes from the high central Southern African plateau downward in the direction of the oceans. The Namib coastal dust sources lay west of the Namibian Great Escarpment.

The direct and semi-direct radiative impact of the widespread BBA plumes has been shown to have an impact on atmospheric
dynamics over the southern Atlantic Ocean (Mallet et al., 2020) as well as on atmospheric circulation and deep convection over the continent at the regional scale (Chaboureau et al., 2022). Chaboureau et al. (2022) have shown that the radiative effects of BBA contribute to (i) the lofting of the smoke plume, (ii) the intensification of deep convection over Central Africa, (iii) the strengthening of the southern African easterly jet (AEJ), and (iv) the weakening of the Benguela low-level jet along the western Namibia coast.

Through their radiative impact at the regional scale, BBAs have the potential to modify low-level atmospheric circulation and hence dust emissions. The downward mixing of the nocturnal low-level jet (LLJ) after sunrise, and the associated downward transfer of momentum, is responsible for most of the high wind speeds occurring during mornings over desert regions in West Africa (Washington and Todd, 2005) and southern Africa (Clements and Washington, 2021). Assessing whether the strong surface winds leading to dust emission are enhanced or weakened as a result of BBA radiative forcing is of importance
since dust uplift potential is a function of the wind speed cubed as well as a threshold wind velocity, i.e. the minimum wind speed initiating the wind erosion. Thereby, the threshold wind velocity is controlled by surface characteristics (Marticorena and Bergametti, 1995). Furthermore, assessing the radiative impact of BBA on the mid-level circulation associated with the transport route of BBA away from the fire-prone regions, such as the rivers of smoke, is also of significance as they can affect marine bio-geochemistry by deposition to the southern and Indian oceans (Ito and Kok, 2017) and are responsible for impairing
air quality as far in Australia (Sinha et al., 2004).

In this study, we take advantage of the unique and comprehensive observational datasets gathered during an episode of dust emission from Etosha pan in the morning of 5 September 2017 to investigate the BBA radiative impact on the near-surface flow responsible for activating the dust sources. This episode was associated with moderate low-level winds. On this day, a river of smoke was also observed along the western coastline of Namibia (Flamant et al., 2022). The datasets were acquired in
the course of the Aerosols, Radiation and Clouds in southern Africa (AEROCLO-sA) field campaign (Formenti et al., 2019)





and consist mainly of space-borne, airborne and ground based remote sensing observations, near-surface in situ observations, as well as high-resolution simulations performed with the mesoscale non-hydrostatic model Meso-NH (Lac et al., 2018).

The paper is organized as follow: Sections 2 and 3 detail the data (observations as well as simulations, respectively) used in this study. In particular, we detail the ensemble simulations designed with and without BBA radiative impact. Section 4 briefly
describes the synoptic situation on of 5 September 2017 associated with the river of smoke over Namibia and the Etosha dust activation episode and associated emission processes based on observations and simulations with BBA radiative impact. In Section 5, we provide an overview of the aerosols, clouds and circulations over the Namibian plateau in connection with a river of smoke, based on regional scale observations and simulations. We evidence the good agreement between a variety of observations and the simulation that include accounts for the radiative impact of BBAs. In Section 6, we detail the impact of
BBA on the lower tropospheric circulation, including dust emissions and BBAs transport in the river of smoke by comparing the two sets of ensemble simulations.

## 2 Observations

### 2.1 Airborne observations

Details of the high-flying French Falcon 20 aircraft for environmental research of Safire (Safire FA20) during AEROCLO-sA
as well as the FA20 payload are provided in Formenti et al. (2019). Operations on 5 September in the morning during the episode of activation of the Etosha pan dust sources are provided in Formenti et al. (2019) and Flamant et al. (2022). The raster flight pattern over Etosha was oriented in such a way that the pan was overflown from east to west along four legs roughly perpendicular to the northeasterly low-level flow forecasted by the European Center for Medium-Range Weather Forecasts (ECMWF), three over Etosha and one dowstream. The timing of the flight was imposed by the necessity of the Safire FA20 to
reach the easternmost cross-low level flow after sunrise (07:00 LT, 05:00 UTC) in order for the airborne instruments to measure data relevant to the understanding of the dust activation processes.

The Safire FA20 took-off from Walvis Bay at 07:36 UTC and flew directly towards the southeastern corner of the Etosha pan and rallied the first of the three cross-pan leg at 08:25 UTC. The Safire FA20 released the first of two dropsondes over Etosha at 08:27 UTC, and a second dropsonde at 08:39 UTC. A third dropsonde was released at 09:06 UTC downstream of the Etosha
pan along the fourth leg of the raster pattern (Fig. 1a). The Safire FA20 then performed a large scale 'inverse L-shape' pattern to document the southwestward dust transport towards the ocean, with a fourth dropsonde being released at 09:20 UTC along the east-west leg of the pattern.

In addition to the dynamics and thermodynamics profiles obtained from dropsondes released by the Safire FA20, the vertical structure of the aerosol layers (incl. BBAs and dust) was obtained from the nadir-pointing airborne lidar LEANDRE Nouvelle
Génération (LNG; Bruneau et al., 2015) flying on the Safire FA20 as well. LNG operates at three wavelengths (355, 532 and 1064 nm), but in the following we shall conduct the analysis of extinction coefficients at 532 nm only. They are retrieved from attenuated backscatter coefficient profiles using a standard lidar inversion method that employs a lidar ratio characteristic of BBA (the dominant aerosol species present over continental Namibia, as discussed later). The lidar ratio value was derived from



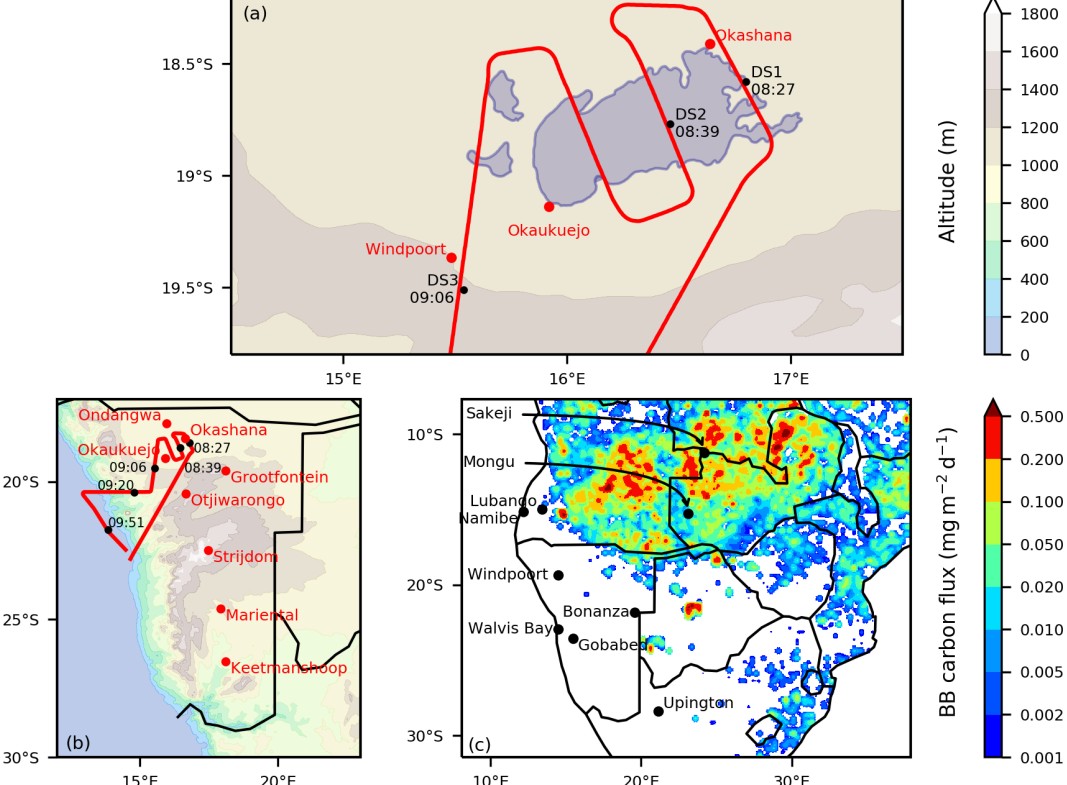

**Figure 1. (a)** Safire FA20 flight plan over the Etosha pan, Namibia, on 5 September 2017 (red thick line). The pan appears as the white area. The location of the three first dropsondes (DS) released from the aircraft are also indicated (DS1 at 08:27 UTC, DS2 at 08:39 UTC and DS3 at 09:06 UTC) **(b)** Full flight plan of the Safire FA20 over Namibia on 5 September 2017 (red thick line). The location of the five dropsondes released from the aircraft are also indicated (incl. DS4 at 09:20 UTC and DS5 at 09:51 UTC) as well as the location of SYNOP stations used in this study. In panels **(a)** and **(b)**, the color shading shows the altitude terrain. The Namibian Great Escarpment is the steep orography feature running along the Atlantic coastline. **(c)** Meso-NH domain. The color shading shows the GFED emission of biomass burning carbon averaged between 1 and 5 September 2017. The black dots indicate the location of the AERONET stations used in this study.

the MPL measurements made in Windpoort on the day of the flight (see Section 4). The vertical resolution of the extinction

coefficient profiles is 30 m, and are averaged over 5 s, yielding a horizontal resolution of 1 km for an aircraft flying at 200 m s$^{-1}$, on average. The extinction coefficient retrievals have an estimated uncertainty of 15 %. The high spatio-temporal resolution of the LNG retrievals enables to discriminate between aerosol layers and clouds, and allows to identify aerosol layers separated by clear air layers which are at least 60–90 m deep.





## 2.2 Ground-based observations

The National Aeronautics and Space Administration (NASA) Aerosol Robotic Network (AERONET; Holben et al., 1998) operates several sun-photometers across southern Africa. Retrievals of aerosol optical depth (AOD) at 532 nm and single-scatter albedo (SSA) at 440 nm from eight stations are used for the verification of the model simulations of these variables at the regional scale (Holben et al., 2018). All level 2.0 AOD and SSA values between 1 and 5 September are used for that purpose. The name and location of the stations are summarized in Table 1 and are shown in Fig. 1c. It is worth noting that
one of AERONET stations is located to the southwest of Etosha Pan, in Windpoort and hence downwind of a prominent dust source.

NASA and the Namibia University of Science and Technology also operate a Micro-Pulse Lidar (MPL) in Windpoort. The MPL system is designed to measure aerosol and cloud vertical structure, and boundary layer heights and is part of the MPL Network Welton et al. (MPLNET; 2001). Here, we use the latest MPLNET processed data (version 3) quick-looks, available
from https://mplnet.gsfc.nasa.gov (Welton et al., 2018); data in Windpoort are not available for download. Extinction coefficient profiles as well as lidar ratio values are not available between 06:29 and 20:15 UTC as the quality of the lidar data is not sufficient for a proper lidar inversion procedure to be conducted. In the following, we focus on particle volume depolarisation ratio profiles. The depolarisation time-height plot is mainly used to analyze the diurnal evolution of the planetary boundary layer (PBL) and its development over Windpoort in the morning of 5 September. Here it is assumed to be representative of
the conditions over the Namibian plateau. The development of the PBL is what leads to the breakdown of the low-level jet in the Etosha region (Clements and Washington, 2021). Together with AERONET retrievals, vertical profiles of the MPL depolarisation ratio are also used to distinguish aerosol layers of biomass burning and dust.

In addition to the passive and active remote sensing instruments measurements, SYNOP stations measurements of 10-m wind speed acquired in the vicinity of Etosha, namely at Ondangwa, Okaukuejo, Grootfontein, and Otjiwarongo as well
as further south over the Namibian plateau have been used (see Table 1 and Fig. 1b). In addition to measurements from the SYNOP stations operated by the Namibian Weather Services, we also used 10-m wind speed measurements from a station operated by SASSCAL (Southern African Science Center for Climate Change and Adaptive Land Management, http://www.sasscalweathernet.org/) in Okashana. Hourly 10-m wind speeds as well as the maximum wind speed measured during each hourly time step (when available) are used to identify the dust emission potential in each location (independent
of whether they are source regions). Wiggs et al. (2022) have estimated the threshold wind velocity to be 7.25 m s$^{-1}$ over Etosha during the dry season. In the following, the 10-m mean and maximum wind speed values measured at the location of the SYNOP stations will systematically be compared to the threshold value of 7 m s$^{-1}$ in order to assess the likelihood of the dust emissions.

## 2.3 Space-borne observations

In this study, we use of AOD retrievals at 550 nm from the NASA Moderate Resolution Imaging Spectroradiometer (MODIS; King et al., 1992). We also make use of extinction coefficient profiles at 1064 nm obtained from the space-borne lidar Cloud-



**Table 1.** Name and position of the SYNOP stations and AERONET stations of interest in this study. The stations are listed from north to south.

| SYNOP stations | | AERONET stations | |
|---|---|---|---|
| Location | Position (longitude, latitude) | Location | Position (longitude, latitude) |
| Ondangwa | 15.97°E, 17.88°S | Mongu (Zambia) | 23.15°E, 15.25°S |
| Okashana | 16.64°E, 18.41°S | Sakeji (Zambia) | 24.31°E, 11.23°S |
| Okaukuejo | 15.92°E, 19.14°S | Namibe (Angola) | 12.18°E, 15.16°S |
| Grootfontein | 18.12°E, 19.60°S | Lubango (Angola) | 13.44°E, 14.96°S |
| Otjiwarongo | 16.67°E, 20.43°S | Windpoort (Namibia) | 15.48°E, 19.37°S |
| Strijdom | 17.47°E, 22.48°S | Bonanza (Namibia) | 19.59°E, 21.83°S |
| Mariental | 17.93°E, 24.60°S | Gobabeb (Namibia) | 15.04°E, 23.56°S |
| Keetmanshoop | 18.12°E, 26.53°S | Upington (South Africa) | 21.16°E, 28.38°S |

Aerosol Transport System (CATS; Yorks et al., 2016) to gather information on the vertical structure of aerosol and cloud layers from an overpass over southern Namibia, Botswana, and Zambia on 5 September 2017, between 22:04 and 22:19 UTC. Details about the space-borne products used in this study can be found in Chazette et al. (2019).

## 3 Modeling

### 3.1 Numerical weather prediction model reanalyses

The synoptic situation associated with the dust source activation episode over Etosha is described by using the atmospheric variables extracted from the Fifth ECMWF Reanalysis (ERA5, Hersbach et al. (2018)). The reanalysis outputs are available every hour on a 0.25° regular grid, on 137 pressure levels, 88 of which are below 20 km and 60 below 5 km (note that only 37

levels are available for direct download). The zonal and meridional wind at 10 m above ground level are used to represent the near-surface wind associated with the dust emission, wind speed at 825 hPa is used to represent the LLJ, geopotential height at 700 hPa is used to account for the mid-tropospheric circulation associated with the semi-permanent continental high over austral Africa, which modulates the BBA transport westwards and southeastwards, and vorticity at 300 hPa is used to describe the dynamics of transient systems. The synoptic evolution on the 5 September 2017 is compared with a 20-year climatology,

computed as the mean from 3 to 7 September over the period 1997-2016.

### 3.2 Meso-NH convection-permitting simulations

The nonhydrostatic mesoscale model Meso-NH (Lac et al., 2018) version 5.4 is used to run the BBRAD (with BBA radiative effects) and NORAD (without BBA radiative effects) simulations. Meso-NH is run over a domain covering both BBA sources over southern Africa and dust emissions over Namibia (Fig. 1c) using a horizontal spacing of 5 km and 64 levels in the vertical

ranging from 60 m above ground to 600 m in the free troposphere. Initial and lateral boundary conditions are taken from the





ECMWF operational analysis. The parameterizations used are the Surface Externalisée (SURFEX) scheme for surface fluxes (Masson et al., 2013), the Rapid Radiative Transfer Model (Mlawer et al., 1997) for longwave radiation, a two-stream scheme for shortwave radiation (Fouquart and Bonnel, 1986), a 1.5-order closure scheme for turbulence (Cuxart et al., 2000), an eddy diffusivity mass-flux scheme for shallow convection (Pergaud et al., 2009), a bulk microphysical scheme for mixed-phase clouds (Pinty and Jabouille, 1998) and a subgrid-scale cloud cover and condensate scheme (Chaboureau and Bechtold, 2005).

The BBA scheme consists of a biomass burning carbon tracer emitted into the first vertical layer of Meso-NH from the daily Global Fire Emissions Database (GFED) version 4 (van der Werf et al., 2017b). The BBA tracer is then mixed by turbulence within the boundary layer and transported by airflows. This simple configuration is sufficient to realistically simulate BBA transport for 2 weeks, as shown by Chaboureau et al. (2022). A mass extinction efficiency of $5.05\,\mathrm{m^2\,g^{-1}}$ at 532 nm and a single-scattering albedo (SSA) of 0.85 are used for BBA as in Mallet et al. (2020) and Chaboureau et al. (2022). The Grini et al. (2006) dust scheme takes into account dust emissions, which are calculated from wind-friction speeds using the Dust Entrainment and Deposition (DEAD) scheme (Zender et al., 2003), and dust transport, dry and wet deposition using the ORganic and Inorganic Log-normal Aerosols Model (ORILAM) (Tulet et al., 2005). It has been extensively validated over the Sahara and the Middle East (e.g., Bou Karam et al., 2009; Lavaysse et al., 2011; Chaboureau et al., 2016; Francis et al., 2017).

For each of the two radiative configurations (BBRAD and NORAD), an ensemble of five members are integrated until 00:00 UTC on 6 September 2017. Their initial time is shifted by 6 h between 00:00 UTC on 31 August 2017 and 00:00 UTC on 1 September 2017. These two ensembles of five members allows us to use the two-tailed Student's $t$ test (at the 95% confidence level) to assess the significance of changes due to the radiative effect of BBA.

## 4 The dust activation episode over Etosha and the Synoptic situation

### 4.1 Synoptic conditions

The synoptic situation leading to the Etosha dust source activation has been described in Formenti et al. (2019) and Flamant et al. (2022), with the former providing more specific details with respect to the dynamics associated with dust emissions over the Namibian plateau. In this section, the relationship between the LLJ evolution and the large scale circulation is described, based on the ERA5 wind fields. In early September, climatological mean large scale circulation over the region is characterised by the presence of a semi-permanent anticyclone dominating the mid-tropospheric circulation above southern Africa with a center over Namibia (Fig. 2a). On 5 September 2017, the center of this feature was displaced eastward over the Indian Ocean just south of Madagascar (Fig. 2b), leaving room to two eastward moving troughs (identified by negative vorticity in the upper troposphere). The elongated easternmost trough over Namibia corresponds to the leftover of the cut-off low observed from 2 to 4 September (Flamant et al., 2022). The interaction between the upper and the mid-tropospheric dynamics led to the development of the temperate tropical trough responsible for the formation of the river of smoke observed to cross Namibia and South Africa from 4 to 6 September (Flamant et al., 2022).

Closer to the surface, the winds at 825 hPa (roughly 2 km amsl) associated with the presence of the semi-permanent anticyclone are generally weak over the western Namibian plateau (where Etosha and Windpoort are located). Stronger winds




over Angola and Zambia to the north are associated with the easterly flow along the northern fringes of the semi-permanent
180 anticyclone (Fig. 2c). On 5 September, the south-eastward displacement of the anticyclone resulted in strong LLJ winds at
07:00 UTC across the southern Africa plateau along its fringes as a result of the large pressure gradient (Fig. 2d). Nearly east-
erly LLJ winds between 12 and 16 m s$^{-1}$ are seen over Etosha and Windpoort. Farther south, northerly LLJ winds even exceed
16 m s$^{-1}$ over the southern Namibian plateau. The LLJ as seen in the 825 hPa ERA5 wind fields on 5 September was maximum
at 07:00 UTC and started to breakdown from 08:00 UTC onwards (not shown).

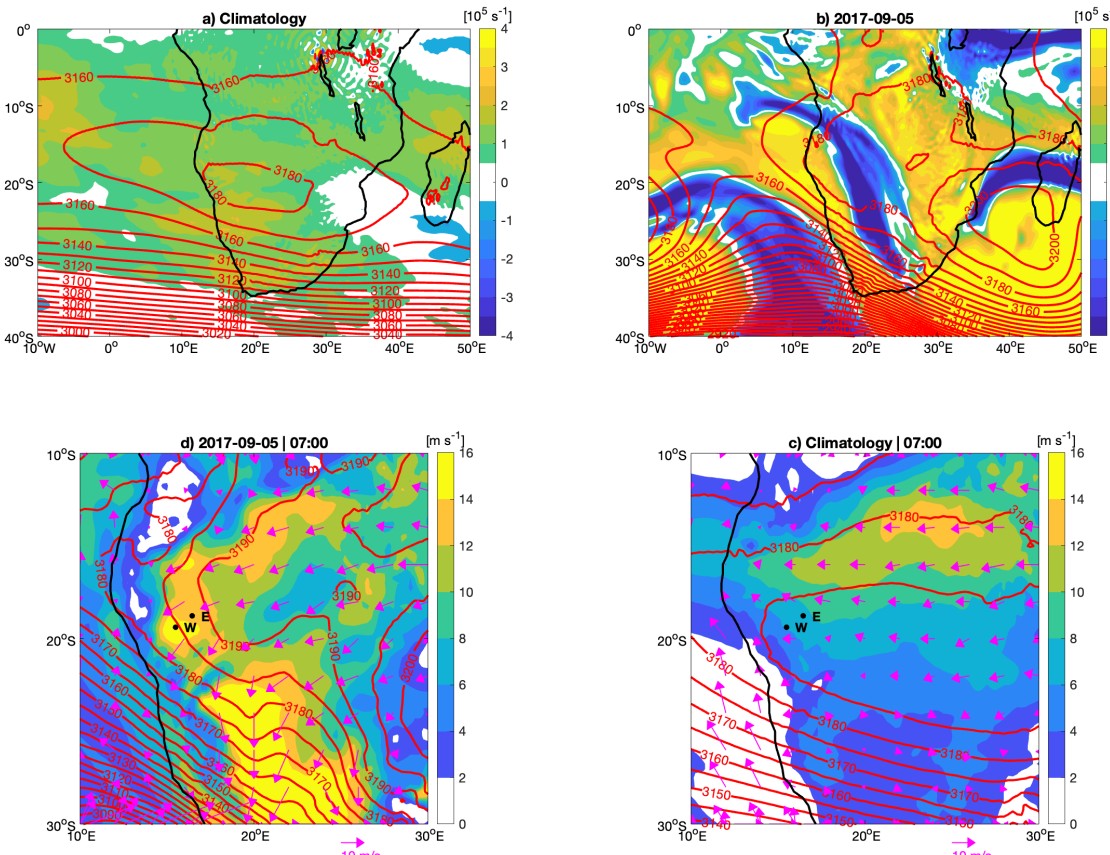

**Figure 2.** Large scale circulation represented by geopotential height at 700 hPa (contours) and relative vorticity at 300 hPa (shadings) for the
1997-2006 climatology (a) and on 5 September 2017 (b). Local scale circulation represented by 10-m wind (vectors), wind speed at 825 hPa
(shadings) and geopotential height at 700 hPa (contours) for the 1997-2006 climatology (c) and on 5 September (d). All ERA5 fields in (a)
and (b) are daily averages. All ERA5 fields in (c) and (d) are at 07:00 UTC.




### 4.2 Dynamical and thermodynamical processes associated with PBL diurnal cycle and their impact on atmospheric composition over the Namibian plateau


The time-height evolution of lidar depolarization ratio profiles from the ground-based lidar on 5 September between 01:00 and 12:00 UTC (Fig. A1a) highlights the presence of deep aerosol layer up to 4 km amsl, as well as the development of a convective PBL from 07:00 UTC onwards as indicated higher depolarization ratio values. The volume depolarization ratios below 3 km

amsl are highest before 06:00 UTC, as well as within the developing convective PBL. However, values are relatively small, as expected from aged BBAs possibly mixed with dust. The PBL development phase is not associated with a noticeable increase in AOD between 07:00 and 12:00 UTC which is in the order of 0.8 at 500 nm (Fig. A1b).

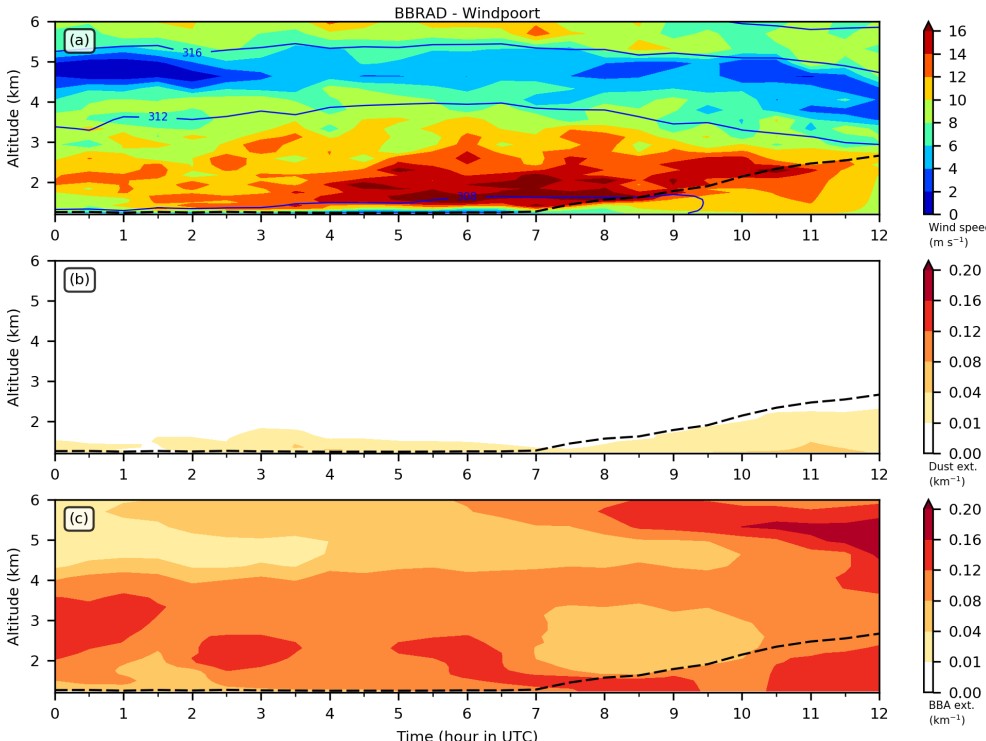

**Figure 3.** Time evolution of **(a)** wind speed, **(b)** dust extinction and **(c)** BBA extinction over Windpoort on the morning of 5 September 2017. The dash black line shows the height of the PBL. In panel **(a)**, the blue contours show the potential temperature every 4 K. Results are shown for the BBRAD member starting at 00:00 UTC on 1 September 2017.

Figure 3a–c shows the time-height cross-sections of wind speed, dust extinction and BBA extinction derived from the BBRAD simulation, for the same period as the MPLNET lidar data shown in Fig. A1a. In the simulation, the convective PBL

starts developing from 07:00 UTC on with a PBL top reaching 2.7 km amsl at 12:00 UTC, as also observed from the lidar measurements. The BBRAD simulation evidences the presence of a low-level jet (LLJ) below 2 km amsl that disappears when the convective PBL develops in the morning (Fig. 3a). The LLJ is persistent throughout the night, as recently observed with



Doppler lidar wind measurements in Etosha pan (Clements and Washington, 2021). The maximum wind speeds in the core of
the LLJ exceed $16\,\mathrm{m\,s^{-1}}$ and is seen above the top of the convective PBL once it has begun to develop. The presence of the

LLJ is also observed in the dropsondes measurements made near Windpoort at 09:06 UTC, i.e. during the development phase
of the convective PBL (Fig. 4i–k). The wind speed maximum is observed at 2 km amsl, and a weaker, but distinct, secondary
wind speed maximum is also observed at 1.5 km amsl (Fig. 4k). The secondary wind speed maximum lies below the top of the
convective PBL (see the potential temperature profile, Fig. 4i) and is likely created by the downward mixing of the LLJ above,
into the PBL. This mechanism is reproduced in BBRAD with two wind speed maxima of equal intensity being simulated, but

at slightly higher altitudes. The fact that the main LLJ feature above the PBL top is underestimated in the BBRAD simulation,
and the lower wind speed maximum is overestimated may be an indication that the downward mixing of the LLJ in the model
is too rapid and too strong. In the BBRAD simulation, the relatively high winds in the lower troposphere are separated from
the mid-tropospheric circulation by a $\sim$ 1 km thick layer of weak winds centered around 4.5 km amsl. This layer of minimum
winds is also seen in the dropsonde data near Windpoort (Fig. 4k).

Regarding the atmospheric composition, it appears that low dust extinction values are simulated in the first levels of the
model (over a relatively thin layer) throughout the night, and that the dust layer deepens as the convective PBL develops due to
vertical mixing (Fig. 3b). A maximum of dust concentration near the surface is seen at 11:00 UTC. The BBA extinction cross-
section highlights the deep BBA layer up to 4 km amsl as in the lidar observations, except after 06:00 UTC when an elevated
BBA layer is also seen above 4 km amsl which eventually merges with the lower one. The BBRAD simulation suggests that

BBA are gradually incorporated in the developing convective PBL and dust emission is seen to be quite limited in Windpoort
as this area is not a dust emission hot spot as Etosha pan is.

Provided that the timing of events is similar in Windpoort and over Etosha, the MPLNET observations and the BBRAD
simulation too confirm that the Safire FA20 overpass of Etosha pan (between 08:25 and 08:51 UTC) occurred during the
early phase of the convective PBL development and downward mixing of the LLJ momentum towards the surface. Recently,

Clements and Washington (2021) evidenced that the breakdown of the LLJ resulted in strong surface winds between 09:00 and
11:00 LT (between 07:00 and 09:00 UTC) over Etosha pan during the months of August and September as result of downward
momentum transport when the strong LLJ winds are mixed in the PBL.

The timing of dust events is now examined over Etosha using the BBRAD simulation. Figure 5a–c shows the time-height
cross-sections of wind speed, dust extinction and BBA extinction for the BBRAD simulation directly over Etosha at the location

of dropsonde DS2 (Fig. 1a). A well defined, but shallower and stronger than in Windpoort, LLJ is simulated above the nocturnal
BL and the subsequently developing convective PBL (Fig. 5a). The region of weak winds, roughly between 4 and 5 km amsl
is also seen here as it is seen in Windpoort. The dropsonde data acquired at 08:39 UTC (Fig. 4e–g), over the pan, evidence
the developing convective PBL with a top at 1.5 km amsl, i.e. just below the LLJ and its associated maximum wind speed of
$16\,\mathrm{m\,s^{-1}}$. In the observations, there is a very small hint of a secondary wind maximum in the PBL (Fig. 4g), as observed later

near Windpoort. This is an indication for the downward mixing of momentum may just have started. The LLJ feature is well
reproduced by the BBRAD simulation, although slightly underestimated. The downward mixing process appears not to have
started in the simulation at the time dropsonde DS2 was released over Etosha pan.



**Figure 4.** Profiles of **(a, e, i, m)** potential temperature, **(b, f, j, n)** relative humidity, **(c, g, k, o)** wind speed from dropsondes released at 08:27, 08:39, 09:06, and 09:20 UTC (blue dots) and simulated in BBRAD (orange) and NORAD (green) at 09:00 UTC on 5 September. Profiles of **(d, h, l, p)** extinction at 532 nm derived from the airborne lidar LNG at the location of the dropsondes, and simulated in BBRAD (orange) and NORAD (green) at 09:00 UTC on 5 September. Results are shown for the BBRAD and NORAD members starting at 00:00 UTC on 1 September 2017.



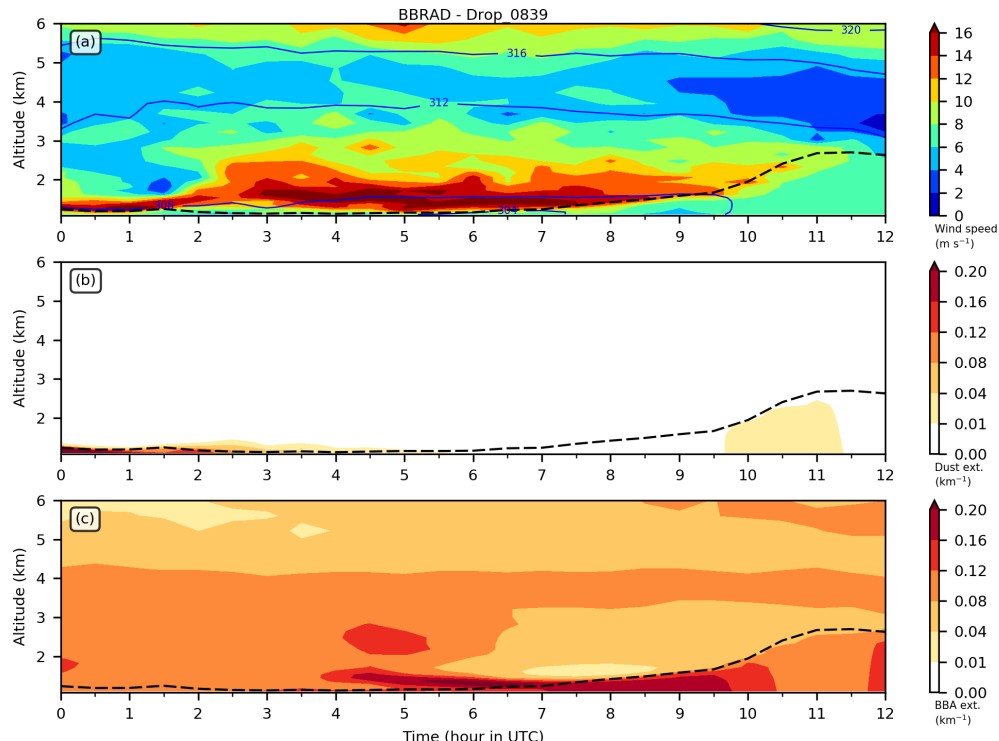

**Figure 5.** As in Fig. 3 but over Etosha pan at the location of DS2 released at 08:39 UTC (16.46° E, 18.77° S).

In the BBRAD simulation, a maximum of dust concentration near the surface is seen at between 10:00 and 11:00 UTC (Fig. 5b), i.e. the time when the strongest surface winds ($\approx$4 m s$^{-1}$) were observed at Okashana and Okaukuejo (see Fig. 6) to

the northeast and the southwest of Etosha pan, respectively. Interestingly, maximum wind speeds in Okashana are just below the threshold velocity of 7 m s$^{-1}$ for dust entrainment until they are exceeded at 12:00 UTC. West of Etosha, the Ondangwa SYNOP station measured the largest wind speeds of up to 7.2 m s$^{-1}$ at 10:00 UTC. In all cases, and consistent with the 10-m wind observations, very little dust emissions are seen in the simulation. On the other hand, BBAs present above the nocturnal BL appear to be incorporated in the developing convective PBL starting at 07:00 UTC, thereby leading to fairly high extinction

values in the PBL, much higher than those associated with the emitted dust. This suggests that the high near surface extinction values simulated below the LLJ in BBRAD are associated with downward mixing of BBA and, to a lesser extent, dust being emitted from the surface of the pan.

## 5 Overview of aerosols, clouds and dynamics over the Namibian plateau in connection with a river of smoke

Airborne lidar extinction at 1064 nm provides a large scale overview of the distribution of aerosols and clouds over and around

Etosha pan and Windpoort (Fig. 7a). The first outstanding feature captured by the airborne lidar measurements is the ubiquitous widespread BBA layer covering the Namibian plateau. BBA observations from this Safire FA20 flight have already been



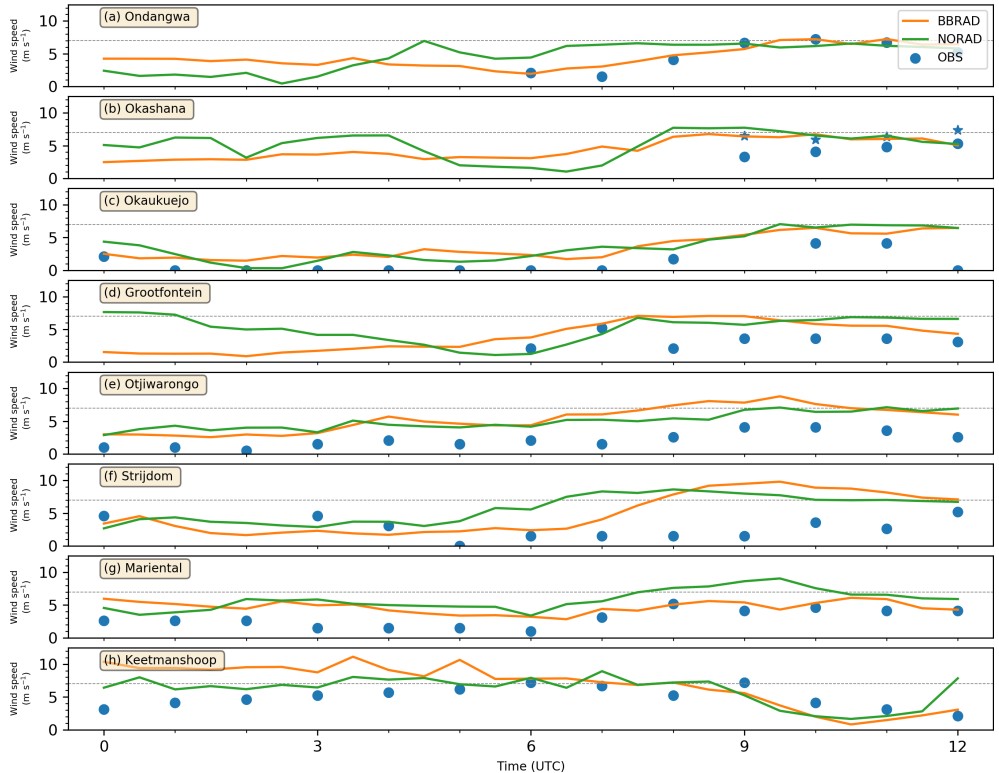

**Figure 6.** Hourly 10-m wind speed (blue dots) measured at the location of the eight stations listed in Table 1, from north to south: **(a)** Ondangwa, **(b)** Okashana, **(c)** Okaukuejo, **(d)** Grootfontein, **(e)** Otjiwarongo, **(f)** Strijdom, **(g)** Mariental, and **(h)** Keetmanshoo. Maximum 10-m wind speeds are shown as blue stars. 10-wind speeds simulated in BBRAD (orange) and NORAD (green) are also shown. The horizontal dotted line represents the $7\,\mathrm{m\,s^{-1}}$ velocity threshold over which dust sources are likely to be activated over the Namibian plateau.

described in Formenti et al. (2019) and Flamant et al. (2022), the later study focus on the processes of formation of a river of smoke along the western coast of Namibia in connection with a mid-level temperate tropical trough and a cut-off low. Flamant et al. (2022) have shown that the interactions between these features had a role in promoting the southeastward transport

of BBA from fire-prone regions in the tropical band to the temperate mid-latitudes and the southwestern Indian Ocean. On 5 September, the river of smoke is observed in the MODIS AOD (Fig. 8a) and simulated in BBRAD (Fig. 8b) in the form of elongated northwest-southeast oriented band of enhanced AOD. The southeasterly BBA transport within the river of smoke is illustrated by the strong wind at 4 km amsl (Fig. 8b). The river of smoke propagated rapidly across southern Africa between 5 and 6 September 2017, under the influence of the fast evolving temperate tropical trough. The vertical extent of the BBA

layer over northern Namibia was controlled by the location of the cut-off low as it favored ascending motion above the BBA layer. In the presence of this feature the top of the BBA layer over northern Namibia reached altitudes above 8 km, i.e. much higher than the average height of the top of the BBA layer over the regions where the smoke comes from (Angola, Zambia, Zimbabwe), as seen in the vertical distribution of aerosols along the CATS track in Fig. 9a. Besides favoring the increase in



**Figure 7.** Time-height cross-sections of extinction at 1064 nm on 5 September 2017 from **(a)** LNG, **(b)** BBRAD and **(c)** NORAD along the red line shown in the inset of the top panel. LNG observations were taken between 07:53 and 09:55 UTC, and Meso-NH simulations are at 09:00 UTC. In panels **(b)** and **(c)**, the white contours show the cloud fraction at 10 % and the yellow contours the dust extinction at 0.01 and 0.05 km$^{-1}$. The red triangles indicate the location of the dropsondes. Results are shown for the BBRAD and NORAD members starting at 00:00 UTC on 1 September 2017.

the BBA layer vertical extent, the ascending motion associated with the cut-off low also promoted the occurrence of mid-level

clouds over northern Namibia. Such clouds are observed in Fig. 8a in the form of missing data along the elongated MODIS-



derived AOD feature, as well as in Fig. 9a (missing CATS data below the high extinction values at altitudes above 6 km amsl is an indication of the presence of these clouds). Large AOD values are also observed and simulated to the east of the river of smoke on 5 September, particularly over Namibia, Angola, Zambia and the Democratic Republic of Congo, from both MODIS observations, as well as AERONET retrievals (Fig. 10). For instance, daily average values of AOD in excess of 1 are

observed Zambia (Sakeji, Fig. 10b), Angola (in Lubango, Fig. 10d) as well Namibia (Windpoort, Fig. 10e), and sometimes up to 1.8 in Angola (Namibe, Fig. 10c). Except in Upington, the AOD simulated in BBRAD are always lower than observed on 5 September. Figure 10 also shows contribution of dust to the total AOD. The BBRAD simulation confirms the lack of dust emission over Angola and Zambia (Fig. 10a-d) and significant emissions (and hence dust AOD) over most of Namibia and northern South Africa prior to 5 September (Figure 10e, g, h). The contribution of dust to the total AOD on 5 September is not

significant in Windpoort and Bonanza (Fig. 10e-f) as opposed to Upington where dust dominates the AOD signal (Fig. 10h). In Gobabeb, dust contribute significantly to the total AOD on 4 September, and this contribution is seen to decrease rapidly after 12:00 UTC on that day (Fig. 10g).

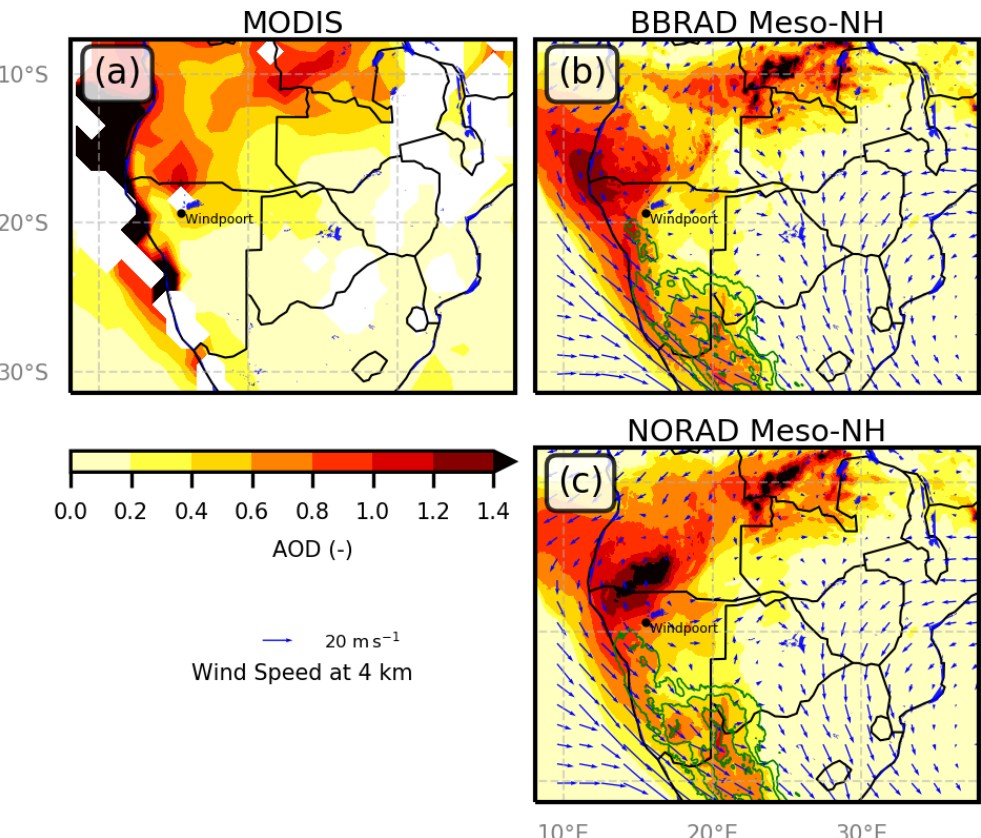

**Figure 8.** AOD at 10:30 UTC 5 September 2017 from **(a)** MODIS, **(b)** BBRAD and **(c)** NORAD. Blue areas represent lakes. In panels **(b)** and **(c)**, the green lines show the AOD due to dust at 0.1, 0.2, 0.4 and 0.6 levels. In panels **(b)** and **(c)**, the blue vectors show the wind at 4 km a.m.s.l..





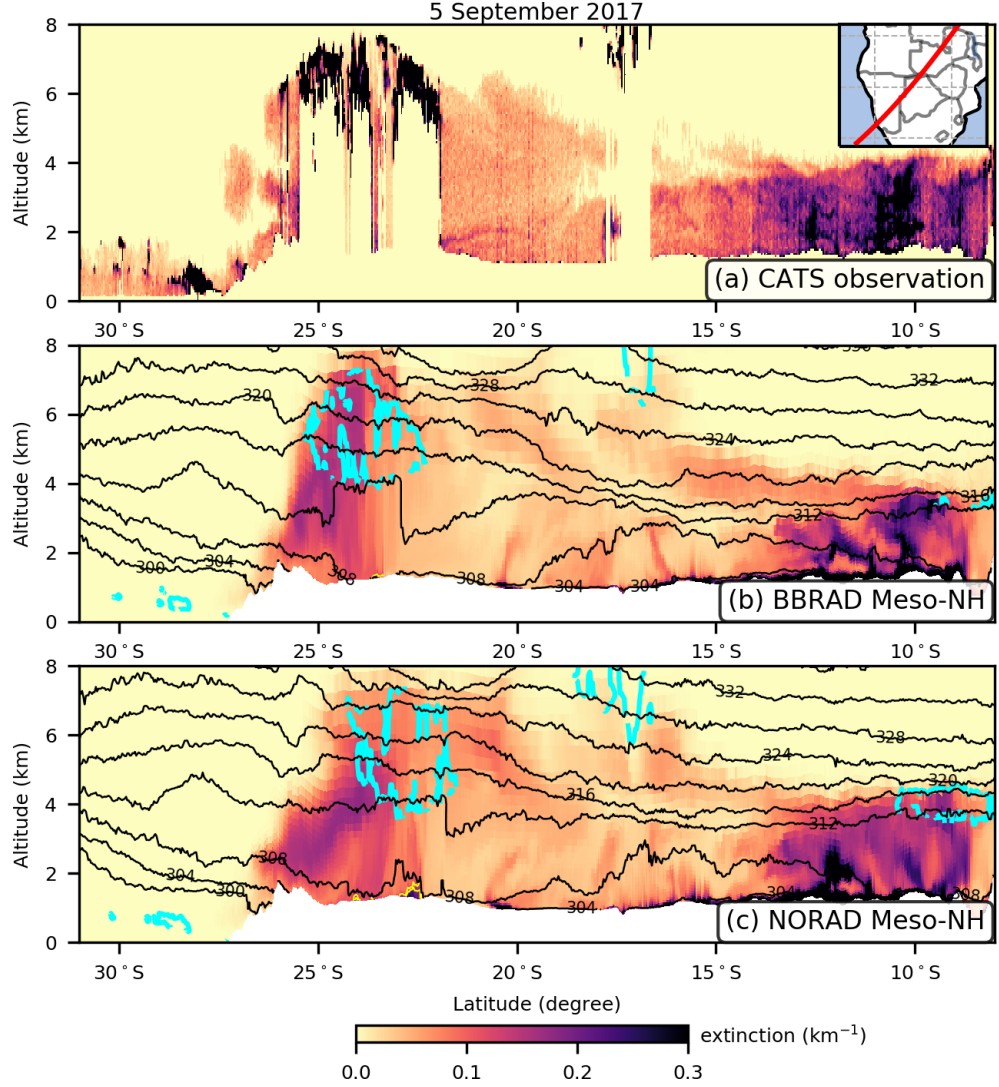

**Figure 9.** Vertical cross-sections of extinction at 1064 nm on 5 September 2017 from **(a)** CATS, **(b)** BBRAD and **(c** NORAD along the red line shown in the inset of the top panel. In panels **(b)** and **(c)**, the black contours show the potential temperature (in K), the cyan contour the cloud fraction at 10 %, and the yellow contours the dust extinction at 0.05 and 0.1 km$^{-1}$. CATS observations were taken between 22:04 and 22:19 UTC, and Meso-NH simulations are at 22:00 UTC. Results are shown for the BBRAD and NORAD members starting at 00:00 UTC on 1 September 2017.

Flamant et al. (2022) also highlighted the deeper BBA layer ($\approx 5$ km) over the Namibian plateau associated with the dy-
namics of the river of smoke, based on both LNG and CATS observations as well as Meso-NH simulations. The altitude of the
top of the BBA layer is observed to reach heights of more than 6–6.5 km amsl, and is at higher altitudes than over the adjacent
Atlantic Ocean (Fig. 7a). In this study, the focus is on the LNG signal from the lower 2 km amsl over Etosha pan and the





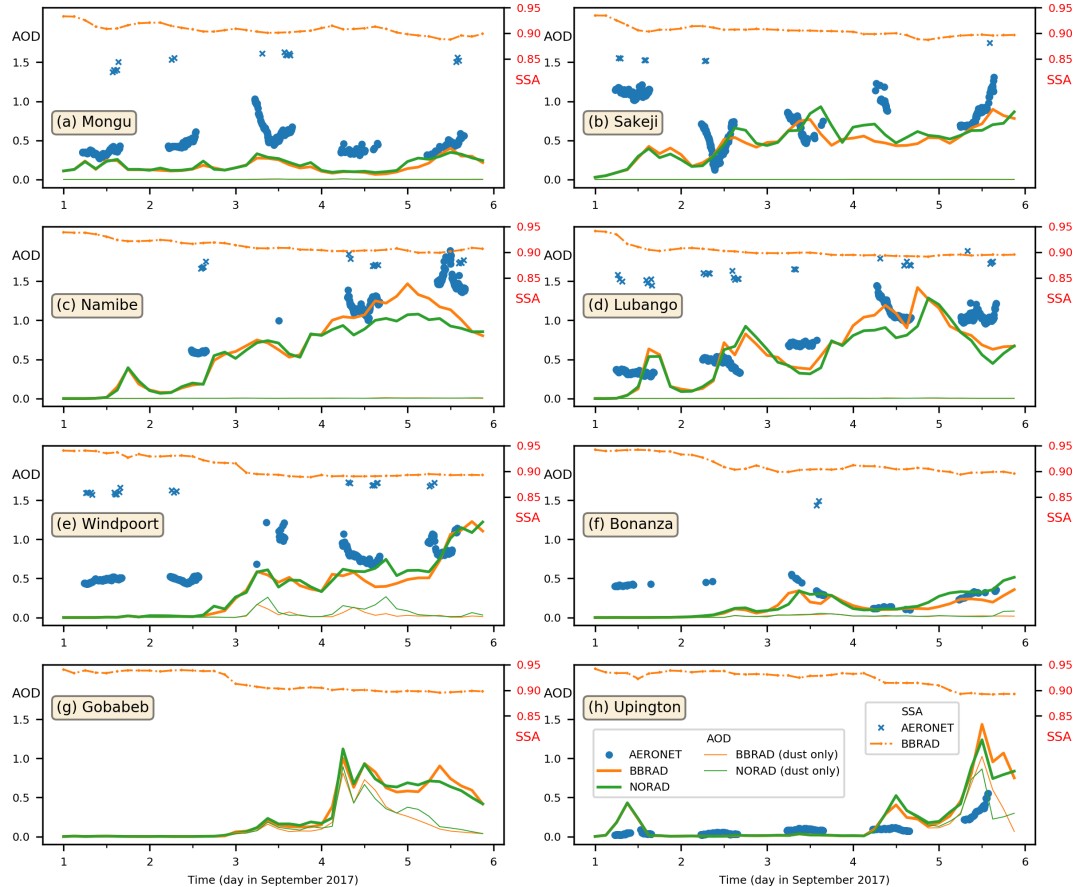

**Figure 10.** Time evolution of AOD at 532 nm between 1 and 5 September 2017 from AERONET (blue), BBRAD (orange) and NORAD (green) at **(a)** Mongu, **(b)** Sakeji, **(c)** Namibe, **(d)** Lubango, **(e)** Windpoort, **(f)** Bonanza, **(g)** Gobabeb and **(h)** Upington. The orange and green thin lines show the AOD due to dust for BBRAD and NORAD simulations, respectively. The blue and orange dotted lines show the SSA at 440 nm for AERONET and BBRAD, respectively. Results are shown for the BBRAD and NORAD members starting at 00:00 UTC on 1 September 2017.

surrounding areas covered by the Safire FA20 flight. Enhanced near surface extinction coefficient values can be observed between 08:30 and 08:40 UTC and around 08:54 UTC while the Safire FA20 crossed the pan. The lower aerosol layer over Etosha was separated from the lofted BBA by a thin layer of lower extinction coefficients indicative of lower aerosol content. In the
Meso-NH BBRAD, enhanced extinction is also seen in roughly the same parts of the flights, with the maximum of extinction over the pan located close to the surface at 08:39 and 08:54 UTC. Comparison between LNG-derived extinction profile at the location of DS2 over Etosha and its simulated counterpart in BBRAD evidences an overestimation in the model, in spite of the fact that the strongest extinction values observed by LNG in the PBL are near the surface (Fig. 4h). As discussed previously, the nearly 300 m deep layer of enhanced extinction above the surface and below the LLJ in BBRAD results from both downward





mixing of BBA and dust being emitted from the surface of the pan. Over Windpoort, the observed and simulated extinction
profiles in the lower 3.5 km amsl are in good agreement, with extinction values being minimum just above the surface (Fig. 4l),
suggesting very small dust emissions and downward mixing of BBA, compared to Etosha.

The enhanced dust extinction coefficients observed and simulated at the beginning and the end of the flight (before 08:00 UTC
and after 09:15 UTC, respectively) are associated with dust emissions from river beds along the flanks of the Namibian plateau,

and are disconnected from the event over Etosha, and likely connected with the low-level dynamics associated with the east-
ward moving river of smoke. For instance, the BBRAD simulation shows dust activation along the Namibian Great Escarpment
as well as over southern Namibia and northwestern south Africa (Fig. 8b). The AOD associated with the activation of Etosha
pan sources is too small to be highlighted in Fig. 8b (related AOD less than 0.1).

As seen in Fig. 7b, dropsonde DS4 was released in a dust emission area activated in BBRAD along the Great Escarpment. The

LLJ observed in this area (Fig. 4o) is at the same altitude amsl as observed near Windpoort, but stronger, while it is simulated
to be lower in BBRAD (and nearly as intense). Large extinction values are simulated below the LLJ with the largest near
surface extinction values simulated amongst the location of the four dropsondes, while strong LNG-derived extinction values
are observed just above the surface, these values also correspond to the largest ones observed at the dropsondes locations with
the airborne lidar (Fig. 4p).

**6   What would the Etosha pan dust emissions and the river of smoke look like in a world without BBA radiative
effect?**

The hourly 10-m wind speed values observed with the SYNOP and SASSCAL stations around Etosha pan are below or just
reach the threshold wind velocity for dust entrainment of 7.25 m s$^{-1}$ estimated over Etosha during the dry season by Wiggs
et al. (2022). Exception made of the Ondangwa SYNOP station, the near surface winds in the broader Etosha region are quite

weak (see Fig. 6) on the morning of 5 September. The instantaneous wind speed measurements closest to the surface of the
pan as measured by dropsondes DS1 and DS2 are 4.2 and 7.8 m s$^{-1}$, respectively, at the eastern edge of the pan and in the
middle of the pan (Fig. 4c, g). The near surface wind speed value in the middle of the pan is just above the threshold provided
by Wiggs et al. (2022). This is likely the main reason why the Etosha dust sources were not activated.

In this conditions, one may wonder to what extent the moderate winds result from a slowing down of the low-level circulation

due to the significant radiative impact of the massive BBA plume over Namibia.

As demonstrated in the previous sections, the Meso-NH BBRAD is realistic compared to a variety of active and passive
remote sensing as well as in situ observations and a suite of dynamical and thermodynamical variables as well as aerosol
properties. Large scale SSA data, gathered from AERONET stations (Fig. 10), evidence that the region of interest is dominated
by absorbing aerosols with SSA ranging between 0.85 and 0.87 on the day of interest, the most absorbing aerosols being

observed in Mongu, Angola (Fig. 10a). Comparisons with the BBRAD simulation show that after five days the aerosols present
over southern Africa are a bit less absorbing than observed, even though the SSA of BBAs was imposed to 0.85. This is due to
the presence of other types of aerosols that BBA and dust as explained in Section 3 and evident at the beginning of the BBRAD





simulation (Fig. 10) when SSA values are significantly larger than 0.85. In all cases, the comparison highlights the fact that BBAs are the dominant aerosols on 5 September over southern Africa.

In order to gain insight into the potential radiative impact of the BBA, we now analyze the dynamics, thermodynamics and atmospheric composition fields extracted from the Meso-NH NORAD ensemble simulation, and compare them to their counterparts obtained from the BBRAD ensemble simulation. First looking at aerosol and cloud distribution at the regional scale:

–   The maximum of AOD in NORAD is displaced to the east as is the elongated northwest-southeast oriented AOD feature
associated with the river of smoke (Fig. 8c vs. 8b). This is consistent with a weaker southern AEJ in NORAD as shown by Chaboureau et al. (2022) when the radiative effects of BBA are omitted in the model,

–   The eastern displacement of the river of smoke in NORAD is also seen when comparing the distribution of cloud fraction, a marker of the strong ascending motion associated with the large scale dynamics, along the CATS track across southern Namibia with the one extracted from the BBRAD simulation (Fig. 9c vs. 9b),

–   The activation of dust sources along the Namibian Great Escarpment is enhanced (Fig. 7c vs. 7b), consistent with the eastward displacement of the river of smoke and the associated low level dynamics. The impact of the low-level dynamics associated with the river of smoke is also evident from the dust source activation over southern Namibia and northwestern South Africa.

The impression left with the evolution of the regional AOD field between BBRAD and NORAD is further quantified in
Fig. 11 for both total AOD (Fig. 11a) and dust-related AOD (DOD, Fig. 11b). The positive anomaly of AOD ($> 0.2$) located along the Namibian coastline is matched by a negative anomaly of similar amplitude ($< -0.2$) the east, over the continent. This AOD anomaly pattern is indeed consistent with a displacement of the river of smoke towards the east in the NORAD simulations. It is worth noting that Windpoort and Etosha are both located in the area of negative AOD anomaly. It is also consistent with the wind speed anomaly at 3.5 km amsl (Fig. 11d) with a positive anomaly ($> 3\,\mathrm{m\,s^{-1}}$) superimposed on
the positive AOD anomaly, and a negative wind speed anomaly ($< -3\,\mathrm{m\ s^{-1}}$) where the negative AOD anomaly is located (Fig. 11c). Both Windpoort and Etosha are located in the area of negative wind speed anomaly. Figure 11e also evidences that the LLJ over the western part of the plateau is much stronger in BBARD than in NORAD which is consistent with the weaker 10-m winds seen in NORAD over Windpoort and Etosha in Fig. 11f at the time when the LLJ starts to break down.

A negative anomaly of dust-AOD ($< -0.1$) is seen along the coastline and west of the Great Escarpment and weaker
positive anomaly (0.05) parallel to it but to the east, over the plateau (Fig. 11b). This is consistent with the surface winds anomaly patterns in the same area (Fig. 11f) with strong wind anomalies along the coastline of Namibia (stronger winds in NORAD). Positive wind speed anomalies are evident southwest of Windpoort and northeast of Etosha, suggesting weaker winds in the NORAD simulation in this area (Fig. 11f), leading to less dust emissions over Etosha in NORAD (Fig. 11c). No significant anomalies in 10-m wind speed are seen around Etosha pan, as corroborated by the comparison with weather
stations of Okashana and Okaukuejo (Fig. 6b, c). Overall, over northern Namibia there are more dust emissions associated



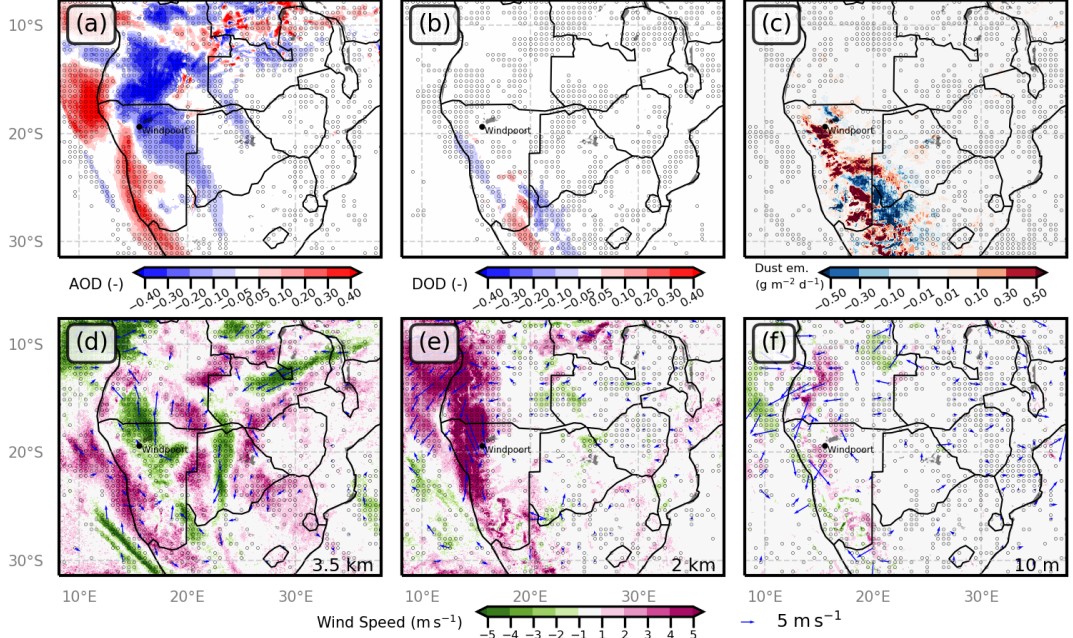

**Figure 11.** Changes between BBRAD and NORAD ensemble means (BBRAD-NORAD) in the **(a)** Total AOD, **(b)** DOD, and **(c)** dust emission. Also shown is wind speed at **(d)** 3.5 km a.m.s.l., **(e)** 2 km a.m.s.l. and **(f)** 10 m a.g.l. Black areas represent lakes. Black open dots indicate where changes in AOD, DOD and wind speed are statistically significant at the 0.05 level. Arrows indicate wind field anomalies when significant at the 0.05 level. Fields are shown at 10:30 UTC 5 September 2017.

with BBRAD. On the other hand, the activation of Namibian coastal sources (or lack there of) is not affected if BBA radiative impacts are not taken into account. Further south, over southern Namibia and northwestern South Africa, a bipolar dust-AOD anomaly pattern (reversed with respect to the one further north) is seen, with positive anomalies ($> 0.2$) to the west and negative anomalies to the east. It is consistent with the dust emission anomaly pattern and likely due to the eastern shift of the river of

smoke in the NORAD simulation and the associated change in low-level dynamics. This is confirmed by the temporal shift of the total and dust-related maximum on 5 September in Upington (Fig. 10h) and also in Bonanza (Fig. 10f), with the AOD peaks occurring in NORAD before the AOD peaks in BBRAD as the river of smoke drifts from west to east during the day (Flamant et al., 2022). This is also corroborated by the simulated 10-m winds at the location of Strijdom and Mariental (Fig. 6f, g) which show an increase of wind speed occurring earlier in NORAD than in BBRAD. However, hourly 10-m wind in these

two locations are much weaker than simulated and below the threshold velocity. Only in Keetmanshoop do we observe hourly 10-m that reach the threshold velocity between 06:00 and 09:00 UTC which are well simulated by both BBRAD and NORAD (Fig. 6h).

     Figure 12 shows the impact of BBA radiative effect on dynamics, thermodynamics and atmospheric composition averaged at 10:30 UTC on 5 September 2017 and between 18 and $20°$ S (a range of latitude including Windpoort and Etosha). The eastern

displacement of the river of smoke is also seen in the BBA extinction cross-sections when comparing Fig. 12a (BBRAD) and



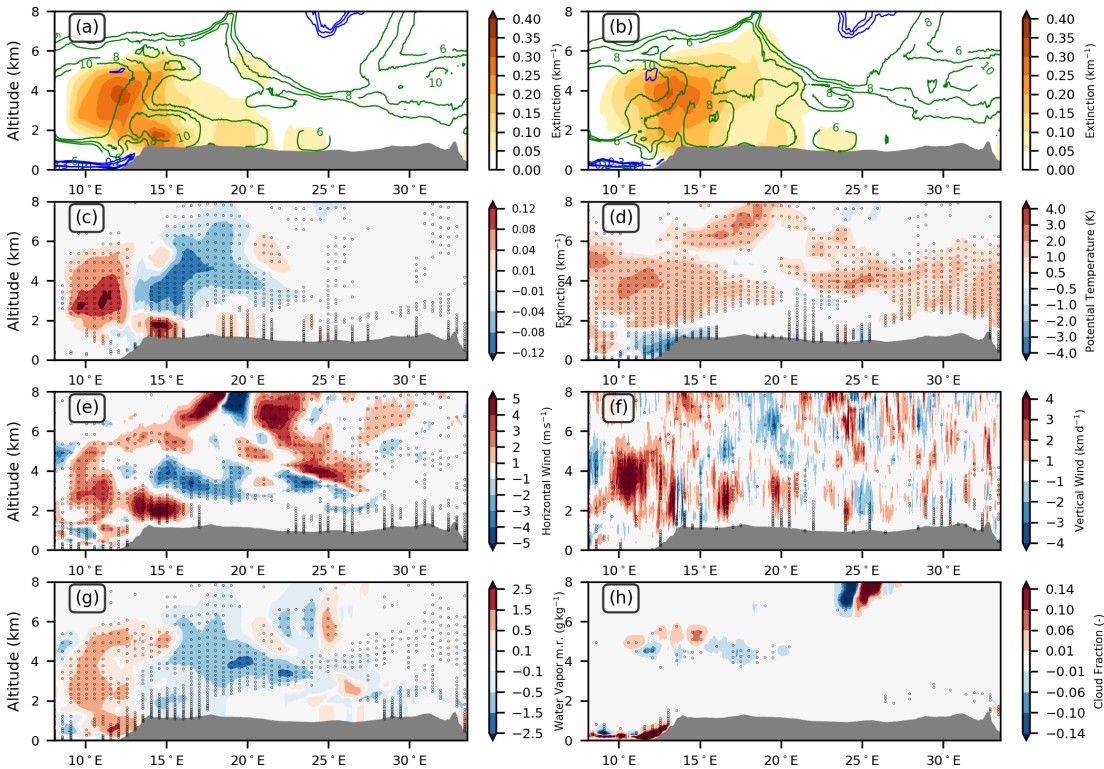

**Figure 12.** (a–b) Extinction (km$^{-1}$, shading), horizontal wind speed (m s$^{-1}$, green contour) and cloud fraction (blue contour) from the (a) BBRAD and (b) NORAD ensemble means. Changes between BBRAD and NORAD (BBRAD-NORAD) in (c) extinction, (d) potential temperature, (e) horizontal wind speed, (f) vertical wind, (g) water vapor mixing ratio and (h) cloud fraction. Black dots indicate where changes are statistically significant at the 0.05 level. Fields are averaged at 10:30 UTC 5 September 2017 and between 18 and 20° S.

12b (NORAD) as well as in Fig. 12c. The larger extinctions in the BBA plume acts to warm the mid-troposphere and cool the lower atmosphere below, this effect being more pronounced to the west (i.e. in BBRAD, Fig. 12d). Warming in the upper part of the BBA plume is seen to be widespread across the Namibian plateau and stronger in BBRAD. Weaker winds at the altitude of the BBA plume are seen over the plateau associated with NORAD while stronger winds farther west (Fig. 12e), consistently

with the finding of Chaboureau et al. (2022) that accounting for BBA radiative effects leads to an acceleration of the southern AEJ. The stronger AEJ in BBRAD leads to increased transport of BBA and higher extinctions over the ocean (Fig. 12a). The differential warming between BBRAD and NORAD in the BBA layer leads to enhanced upward vertical motion over Etosha (Fig. 12f) as well as a drier mid-troposphere (Fig. 12g) and increased cloud fraction (Fig. 12h) to the east consistent with the eastern displacement of the river of smoke in NORAD.

Over Etosha, the convective PBL that develops in NORAD is deeper than in BBRAD (Fig. 13a), as the result of the enhanced vertical motion seen in Fig. 12f. The LLJ above is also weaker than in BBRAD. The weaker LLJ and deeper PBL lead to weaker near surface winds thereby explaining why there are no dust emissions over Etosha in NORAD (Fig. 13b). As in BBRAD, the



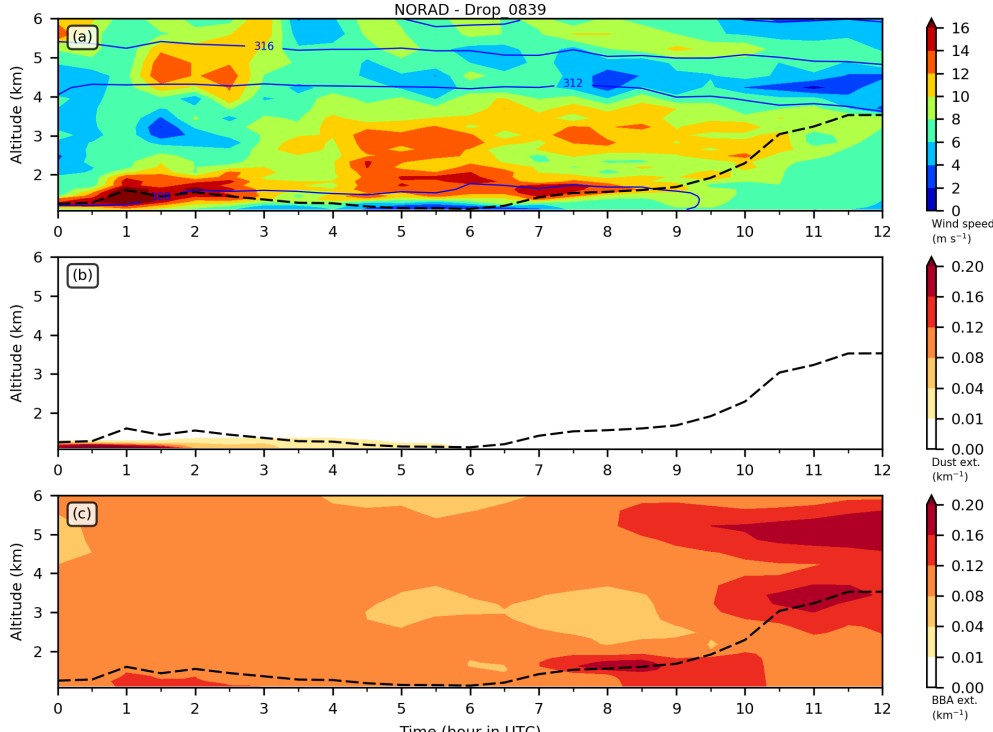

**Figure 13.** As in Fig. 5 but for the NORAD member starting at 00:00 UTC on 1 September 2017.

main contribution to the near surface extinction is linked to the incorporation of BBA in the developing PBL, even though the mixing appears at a later stage in NORAD (compare Fig. 13c and Fig. 5c). The weaker LLJ over Etosha in NORAD is also

seen in Fig. 4 when compared to the dropsonde and BBRAD wind profiles and the wind anomaly at 2 km amsl in Fig. 11e. The elevated maximum of extinction in the NORAD simulation is located above the wind speed maximum of the LLJ and hence is likely related to transport rather than downward mixing of BBA as on BBRAD. The region of weak winds between 4 and 5 km amsl seen in BBRAD is no longer seen in NORAD (Fig. 13b). A weaker LLJ and a deeper developing convective PBL are also seen in NORAD in Windpoort (Fig. 14a). Dust emissions occur in NORAD starting at 09:00 UTC, while dust appears

to be advected before that time above the nocturnal PBL before being incorporated in the developing convection PBL after 07:00 UTC (Fig. 14b). Here also, large amounts of BBA are being incorporated in the developing convective PBL, with the largest extinction values seen around 12:00 UTC in NORAD (Fig. 14c).

  The distribution of clouds and aerosols simulated in NORAD over the plateau (Fig. 7c) reveal more moist conditions in the upper part of the BBA layer with clouds forming within the BBA layer, with cloud bases as low as 4 km amsl. In the lower

atmosphere, dust-related extinction is seen to be mixed to greater heights compared to BBRAD, which is consistent with the deeper developing convective PBL, but dust emissions are not enhanced in the NORAD simulation as seen at the location of dropsondes DS2, DS3 and DS4 in Fig. 4 when comparing extinction coefficients profiles. Furthermore, enhanced extinction



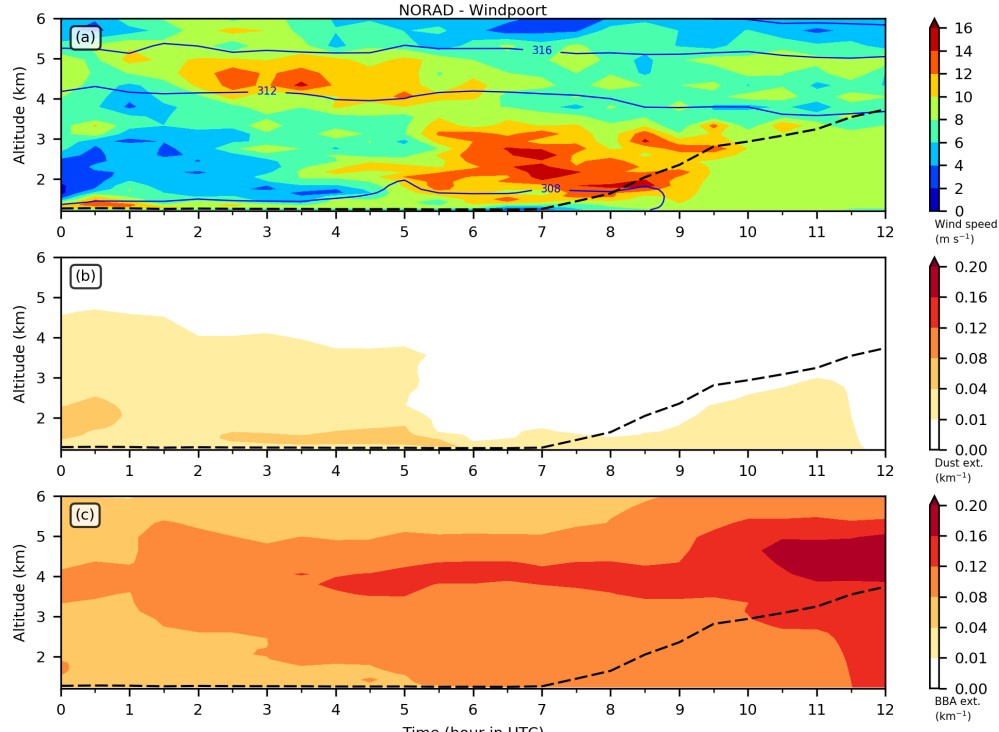

**Figure 14.** As in Fig. 3 but for the NORAD member starting at 00:00 UTC on 1 September 2017.

values at the location of dropsonde DS2 in NORAD compared to BBRAD (Fig. 7c vs. 7b) is due to enhanced downward mixing of BBAs in the convective PBL and their near surface accumulation, not to enhanced dust emissions.

# 7 Conclusions

In this study, we address the radiative impact off BBA on low-level atmospheric circulation (mainly below 5 km amsl) over southern Africa using twin ensemble simulations made with the Meso-NH mesoscale model, one including the direct and semi-direct radiative effects of aerosols (BBRAD) and one in which these effects are not included (NORAD). Our objective is to get insights into the radiative impact of a wide-spread BBA layer on (i) dynamical processes at small scale affecting dust emissions, typically over dust hot spots such as Etosha pan, and (ii) the transport patterns of BBA associated with a river of smoke present over western Namibia on 5 September 2017.

Comparison with airborne and ground-based in situ and remote sensing observations, show that the BBRAD simulation is realistic in reproducing several key features of the local- and large-scale dynamics, thermodynamics and atmospheric composition fields. For instance, simulations and observations confirmed the presence of a LLJ over Etosha, the weak dust emissions (likely related to the fact that near surface winds were just above the threshold wind velocity of $7.25\,\mathrm{m\,s^{-1}}$), as well as the downward mixing of BBA in the developing convective PBL over Etosha. The combination of these datasets also confirm that



the timing of the Etosha pan overpass by the Safire FA20 was well designed to capture morning emissions over this prominent dust source. The breakdown of the LLJ resulting in strong surface winds is simulated between 09:00 and 11:00 LT (between 07:00 and 09:00 UTC) as also evidenced by Clements and Washington (2021), a period during which the Safire FA20 acquired
remote sensing observations and released dropsondes over Etosha pan. The combination of observations and BBRAD simulations also evidence that the LLJ that is present over the plateau west of Etosha pan and that similar dynamical processes occurred there, i.e. developing convective PBL and downward mixing of BBA in the PBL, and that the extinction coefficient in the PBL, which is the major lidar observable, is largely dominated by BBA. The BBRAD ensemble simulation also captures the main dynamical feature present over western Namibia on 5 September, namely the river of smoke described in details by
Flamant et al. (2022). The BBA and cloud distribution associated with this feature, controlled in large part by the presence of a cut-off low and a temperate tropical trough is well represented in the simulation, with the deepening of the BBA layer (its top reaching 8 km amsl) and the presence of mid-level clouds near the top of the BBA layer in the region where the cut-off low exercises its dynamical control.

Comparison between the NORAD simulations and observations evidence that the former is much less realistic than the
BBRAD simulation in representing the key observed features. For instance, not accounting for the BBA radiative impact does not increase near surface winds over Etosha during the LLJ breakdown period as in BBRAD because: i) the LLJ is too weak and ii) the convective PBL is too deep compared to observations. The deeper convective PBL over Etosha and surrounding instrumented sites (such as Windpoort) is related to the enhanced anomalous vertical motion caused by the eastern displacement of the river of smoke in NORAD. This eastern displacement is related to weak southerly AEJ in NORAD compared to BBRAD,
as already shown by Chaboureau et al. (2022). Both BBRAD and NORAD simulations provide clear evidence that the enhanced near surface extinction coefficient values detected from airborne lidar observations over Etosha are related to the downward mixing of BBA in the developing convective boundary layer rather that dust being emitted as a result of the LLJ breakdown after sunrise.

One key finding from this study is that the radiative impact of BBA building up over a period of 5 days in the Meso-NH
simulations can lead to significantly different circulations at low and mid-levels, thereby affecting dust emissions over southern Namibia and northwestern South Africa as well as the transport of BBA. Neglecting the radiative impact of BBA for instance acts to displace the river of smoke towards the east. As these features have an impact on air quality in southern Africa, but also in remote locations such as La Réunion and Australia, it is important the positions of these "BBA transport highways" across the southern Indian Ocean are correctly forecasted. This implies that the radiative impact of BBA has to be realistically
accounted for in numerical weather prediction models to efficiently forecast such episodes. In addition, both radiative active aerosols, mineral dust and BBA, need to be represented accurately in order to simulate the atmospheric dust cycle.

## Appendix A: Particle volume depolarisation ratio measurements from the MPLNET lidar in Windpoort, Namibia

Particle volume depolarisation ratio (VDR) profiles derived from MPLNET lidar measurements in Windpoort are useful to distinguish between aerosol types. It is an intensive parameter, meaning that it depends only on the nature of the aerosol, not



on its concentration or amount (Burton et al., 2012). As such, particle VDR varies according to aerosol type. At first order, it is useful to differentiate spherical aerosol from non-spherical one (Shimizu et al., 2004). However, to some extent, VDR also varies with relative humidity for hygroscopic aerosols (Sassen, 2000). Measurements aerosol VDR values from ground-based Raman lidar have proved to be useful for the separation of aerosol types, including pure dust and biomass burning mixed with dust (Groß et al., 2011). Aged biomass burning and volcanic aerosols can also exhibit some depolarization (Sassen, 2008), but

with much smaller values than for dust or dust mixture. Particle VDR at 532 nm in the range 30–35 % are characteristic of dust from the Sahara desert (Liu et al., 2008), while smaller, yet significant, values of about 20–35 % are often observed for mixtures of dust with other species (Heese and Wiegner, 2008). Biomass burning aerosols are relatively small, spherical particles that produce low VDR (Cattrall et al., 2005), with elevated smoke layer characterized by slightly higher values (8—10 %) than for fresh smoke (<2—5 %) and for more aged smoke (3—8 %).

*Data availability.*    The LNG lidar data is available via the digital object identifier (DOI) https://doi.org/10.6096/AEROCLO.1774 (Flamant, 2018) and the dropsondes data via DOI https://doi.org/10.6096/AEROCLO.1777 (Perrin and Etienne, 2019). The AERONET data were downloaded from the NASA AERONET website (http://aeronet.gsfc.nasa.gov/, last access: 1 September 2023; NASA, 2023a), the MODIS data from the Giovanni web portal (http://disc.sci.gsfc.nasa.gov/giovanni/, last access: 1 September 2023; NASA, 2023b) and the CATS data from ICARE (https://www.icare.univ-lille.fr/, last access: 1 September 2023; AERIS/ICARE Data and Services Center, 2023).

*Financial support.*    The AEROCLO-sA project was supported by the French National Research Agency under grant agreement n° ANR-15-CE01-0014-01, the CNRS-INSU national programs LEFE (Les Enveloppes Fluides et l'Environnement) and PNTS (Programme National de Télédetection Spatiale (grant n° PNTS-2016-14), the French National Agency for Space Studies (CNES), and the South African National Research Foundation (NRF) under grant UID 105958. The research leading to these results has received funding from the European Union's 7th Framework Programme (FP7/2014-2018) under EUFAR2 contract n°312609

*Author contributions.*    CF and JPC conducted the data analysis and prepared the paper with contributions from all co-authors, MG, PF and KS. In addition JPC performed the Meso-NH simulations and produced all the figures, except those related to ERA5 which were produced by MG.

*Competing interests.*    P. Formenti is guest editor of the "New observations and related modelling studies of the aerosol–cloud–climate system in the Southeast Atlantic and southern Africa regions (ACP/AMT inter-journal SI)". The remaining authors declare that they have no conflicts
of interest.





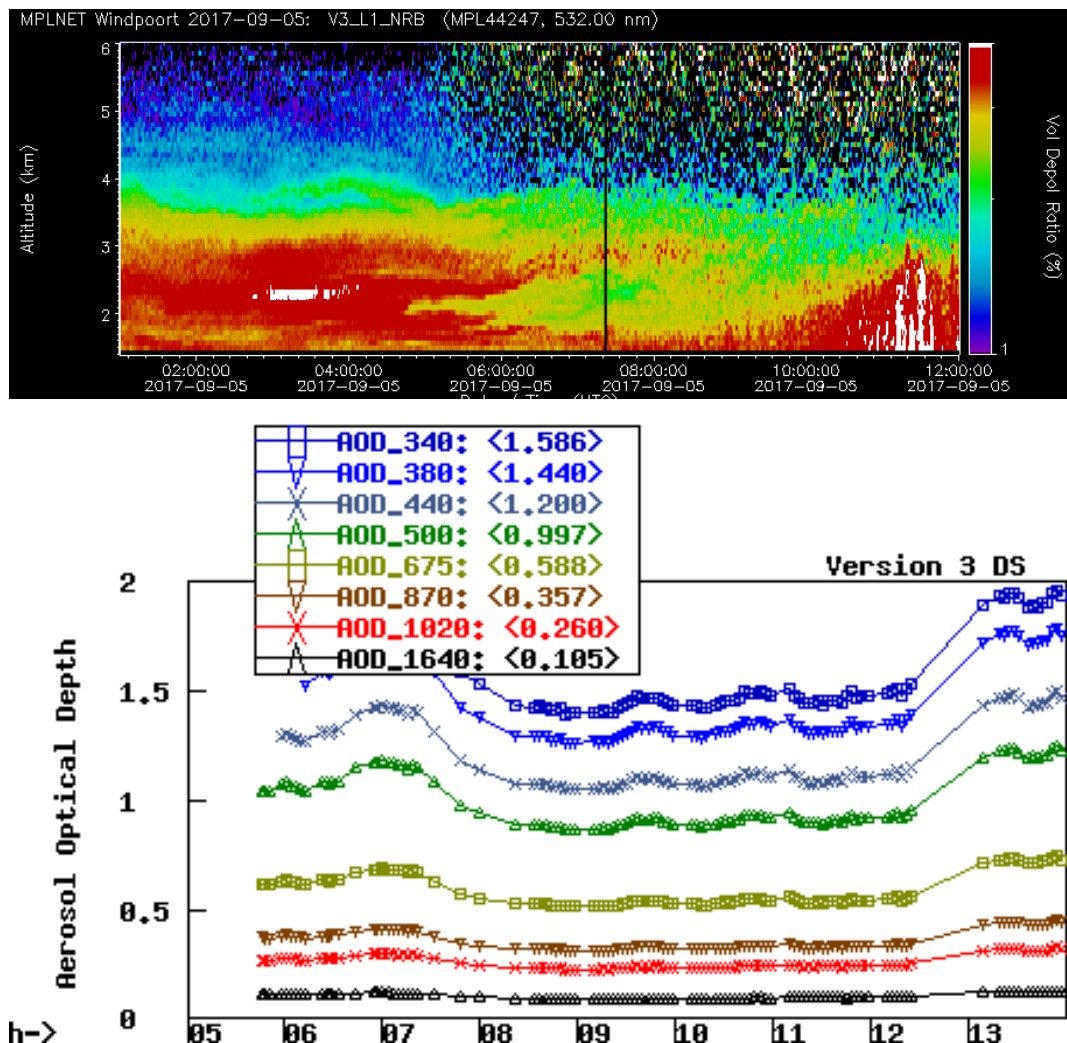

**Figure A1.** (top) Time-height cross-sections of MPLNET lidar particle volume depolarisation ratio and (bottom) time evolution of AERONET-derived AOD in Windpoort on 5 September 2017.

*Special issue statement.* This article is part of the special issue "New observations and related modelling studies of the aerosol–cloud–climate system in the Southeast Atlantic and southern Africa regions (ACP/AMT inter-journal SI)". It is not associated with a conference

*Acknowledgements.* Computer resources for running Meso-NH were allocated by GENCI through Project 90569. Airborne data were obtained using the FA20 aircraft managed by Safire, the French facility for airborne research, an infrastructure of the French National Center for
Scientific Research (CNRS), Météo-France and the French space agency (Centre National d'Etudes Spatiales – CNES). The AEROCLO-sA database and its access are maintained by the French national center for Atmospheric data and services AERIS. We thank the PIs and Co-Is



Margarida Fernandes-Ventura, Pawan Gupta, Brent Holben, Elena Lind, Gillian Maggs-Kolling, Kaleb Negussie, Stuart Piketh, and their staff for establishing and maintaining the AERONET sites used in this investigation, as well as Judd Welton and Nichola Knox for establishing and maintaining the MPLNET lidar site in Windpoort. The authors would also like to thank Larry Oolman, University of Wyoming, for providing the Namibian SYNOP data. The AEROCLO-sA project was supported by the French National Research Agency under grant agreement no. ANR-15-CE01-0014-01, the French national program LEFE/INSU, the Programme national de Télédetection Spatiale (PNTS) under grant no. PNTS-2016-14, the French National Agency for Space Studies (CNES), and the South African National Research Foundation (NRF) under grant UID 105958. The research leading to these results has received funding from the European Union's 7th Framework Programme (FP7/2014-2018) under EUFAR2 contract no. 312609. Airborne data were obtained using the aircraft managed by Safire, the French facility for airborne research, an infrastructure of the French National Center for Scientific Research (CNRS), Météo-France, and the French National Center for Space Studies (CNES). The strong diplomatic assistance of the French Embassy in Namibia, the administrative support of the Service Partnership and Valorisation of the Regional Delegation of the Paris–Villejuif region of the CNRS, and the cooperation of the Namibian National Commission on Research, Science and Technology (NCRST) were invaluable to make the project happen. The support of the aviation authorities is acknowledged. The AEROCLO-sA project would not have been successful without the endless efforts of all the research scientists and engineers involved in its preparation, often behind the scenes. Their support and enthusiasm are sincerely appreciated.



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
