# Peer review of "The radiative impact of biomass burning aerosols on dust emissions over Namibia and the long-range transport of smoke observed during AEROCLO-sA"

_EGUsphere, 2023_

## Referee Comment (RC1)

**Review of "The radiative impact of biomass burning aerosols on dust emissions over Namibia and the long-range transport of smoke observed during AEROCLO-sA" by Cyrille Flamant et al.**

This study investigated the radiative impact of biomass burning aerosols on local and regional atmospheric circulation patterns using a mesoscale non-hydrostatic model. The authors first concluded that their Meso-NH simulations can reasonably reproduce key features of dynamics, thermodynamics and atmospheric composition fields as seen in in-situ and remote sensing observations during a dust emission episode on Sept. 5, 2017. By comparing simulations with and without BBA radiative effects, the authors found that omitting the radiative effects of BBA results in weaker AEJ-S, weaker LLJ, deeper convective PBL, and an eastward shift of the smoke plume.

I found the methodology and the scientific conclusions of this study sound and solid. This study is no doubt publishable with profound implications for air quality and dust cycle in the southern hemisphere. The manuscript is in general well written and easy to follow, however, I do find places where lengthy sentences demand several readthroughs to understand. I believe the readers would appreciate some extra work on these lengthy sentences.

I recommend publish after minor revision.

**General comments:**

First, I am impressed by the capability of Meso-NH in reproducing BBA spatiotemporal distributions and atmospheric circulation patterns during the 5-day time period. I have a few questions regarding the setup of the simulations:

> 1) is there any nudging or re-initialization of the simulation based on ERA5 during the course of the simulation? Is the model free-running for the whole time period since initialization?

> 2) what is the rationale to create ensemble members by shifting initialization time? have you considered perturbing parameters related to BBA, e.g., SSA?

> 3) what exactly is the vertical resolution? And where is the top of your domain (i.e., how do you define "*600m in the free troposphere*")?

> 4) how is "NORAD" achieved? by turning off SW calculations related to BBA in RRTM? How is the semi-direct effect (those related to cloud adjustments) excluded?

I think these details could be added to the main text to help readers better understand the setup of the modeling framework.

Second, I strongly recommend adding a schematic diagram to the manuscript, illustrating the complex pathways through which radiative effects of BBA modulate local and regional atmospheric circulation patterns and the distribution of smoke and dust emissions in southern Africa. When I read through section 6, I had to constantly flip through figures to build such a

mechanistic diagram in my mind, I believe such a diagram in the main text would benefit the manuscript in profound ways.

Other than these two points, I only have a few minor comments and suggestions that I encourage the authors to consider while revising the manuscript.

**Minor comments/suggestions:**

Line 34-39, additional reference for the importance of BBA direct and semi-direct effects: Diamond et al. (2022) pointed out the critical role of accounting for smoke diabatic heating that reduces the free-tropospheric subsidence in reproducing the observed low-cloud cover over the SE Atlantic with a regional model.

Section 4.1, Zhang & Zuidema (2021) also documented the important role of synoptic patterns in governing smoke transport over the oceanic region during the month of September, such that a mid-latitude intrusion pattern (their Fig. 8c, very similar to Fig. 2b in this manuscript) is often associated with less smoky conditions in the remote SE Atlantic, consistent with reduced transport of smoke by a weaker AEJ-S. They show that this synoptic pattern also strongly affects the extent of the marine stratocumulus deck.

Fig. 2, please indicate what "e" and "w" represent in the caption, please check the orientation of a)-d) in this figure.

Fig. 10, there is no blue dotted line on the figure, you meant blue Xs, right?

Line 296-299, this sentence is lengthy and hard to understand, I recommend rewording.

The section titles in general seem a bit long to me, I think more concise and shorter section titles would read and look better.

Line 311, not sure if "*realistic*" is the right word here, I am convinced that the model captured the key features, but I feel "*realistic*" is perhaps too strong here.

Line 330-331, not clear which simulation set you're referring to. "…*is enhanced*" in which simulation? "*… associated low-level dynamics*" could you be more specific? Weaker or stronger LLJ?

Line 355-358, I do not see the temporal shift in Fig. 10f and h, to me, the orange and green curves pretty much track each other at Upington.

Line 371-374, not clear, please indicate "enhanced upward motion and drier mid-troposphere" in which simulation set? The cloud fraction increase is also not clear to me, I see both +ve and -ve cloud fraction changes, are you referring to the signal around 15 E or 25 E?

Line 375-376, my interpretation of the above sentence and Fig. 12f is that upward motion in BBRAD is enhanced due to differential warming, but here, you indicate NORAD has enhanced

vertical motion. I am confused. Also, could you be more specific about how the strength of updraft (vertical motion), LLJ and surface wind is dynamically related? (again, a schematic would help a lot).

**References**

Diamond, M. S., Saide, P. E., Zuidema, P., Ackerman, A. S., Doherty, S. J., Fridlind, A. M., Gordon, H., Howes, C., Kazil, J., Yamaguchi, T., Zhang, J., Feingold, G., and Wood, R.: Cloud adjustments from large-scale smoke–circulation interactions strongly modulate the southeastern Atlantic stratocumulus-to-cumulus transition, Atmos. Chem. Phys., 22, 12113–12151, https://doi.org/10.5194/acp-22-12113-2022, 2022.

Zhang, J. and Zuidema, P.: Sunlight-absorbing aerosol amplifies the seasonal cycle in low-cloud fraction over the southeast Atlantic, Atmos. Chem. Phys., 21, 11179–11199, https://doi.org/10.5194/acp-21-11179-2021, 2021.

---

## Referee Comment (RC2)

**Review comments: egusphere-2023-2371**

**General comments:**

The authors of the study have conducted a comprehensive investigation into the radiative effects of biomass burning aerosols on atmospheric circulation at low and mid-levels over southern Africa. The study, contextualized within the AEROCLO-sA field campaign on September 5, 2017, utilized both in situ and remote sensing observations, alongside sets of ensemble simulations using the Meso-NH mesoscale model.

Their findings highlight that the simulation incorporating BBA radiative effects was shown to represent regional dynamics, thermodynamics, and compositional features more convincingly. In contrast, simulations neglecting these effects presented unrealistic dynamics, such as a weakly formed LLJ and inaccurately represented mid-level dynamics critical for the transport of BBAs. Specifically, the absence of BBA radiative effects resulted in discrepancies such as an inadequately established LLJ overnight and a convective planetary boundary layer (PBL) that was too deep in comparison with observational data. Additionally, the research connected the enhanced near-surface extinction coefficient values observed over Etosha with the downward mixing of BBAs in the convective boundary layer, rather than dust emission due to LLJ breakdown after sunrise.

The study concludes with a clear and important message that for accurate forecasting of dust emissions in Namibia, it is essential to consider the radiative effects of biomass burning aerosols, underscoring their significant influence on atmospheric behavior. However, I have some reservations about some aspects. It is good that this is a very detailed paper, but there is lot of bouncing backwards and forwards between figures in the results section. I would suggest removing some redundant details in the paper and streamlining the text. The details of the model configurations are somewhat missing. I recommend a minor revision before publication.

**Specific comments:**

**Comment 1: Page 3, line 59** "In particular, we detail the ensemble simulations designed with and without BBA radiative impact…"

Please provide a short overview of 'ensemble simulation' of what? In the introduction emphasize the importance of model simulations in the context of the current topic of interest. Include details that underscore the synergistic method combining both measurement and modeling techniques.

**Comment 2:** Could you provide a very short background information on the overall data collection during the campaign and explain the reasons for selecting this particular day for the current analysis? This point seems somewhat obscured in the text and is not clearly articulated.

**Comment 3: Page 5, line 120** "Wiggs et al. (2022) have estimated the threshold wind velocity to be 7.25 ms$^{-1}$ over Etosha during the dry season."

Can you explain what is meant by 'threshold wind velocity'?

**Comment 4: Page 6, Section 3.2** Please add the details of the model configuration: boundary layer scheme, aerosol scheme, etc. Is the model's horizontal resolution set at 5 x 5 km, and does it reach a maximum altitude of 600 m? Please provide more details about BBRAD and NORAD to help the readers understand their relevance better.

**Comment 5: Page 7, line 169-171** "In early September, climatological mean…"

Could you specify the exact time period?

**Comment 6: Page 8, Line 179-180** "over Angola and Zambia to the north are associated with the easterly flow along the northern fringes of the semi-permanent anticyclone (Fig. 2c)."

Is this the average at 07:00 UTC?

**Comment 7: Page 8, Figure 2.** Are 'E' and 'W' in the lower two panels referring to Etosha and Windpoort, respectively? Please mention this in the captions.

**Comment 8: Page 9, Figure 3** Figure caption
Is this the result from one of the BBRAD ensemble members or the average of all the members?
Are the values from the model hourly? If the model outputs are on an hourly basis, could the infrequent recording of the data introduce uncertainties when representing phenomena on a sub-hourly scale?

**Comment 9: Page 13 Line 252-254** "The southeasterly BBA transport within the river of smoke is illustrated by the strong wind at 4 km amsl (Fig. 8b). The river of smoke propagated rapidly across southern Africa between 5 and 6 September 2017, under the influence of the fast evolving temperate tropical trough."

The first sentence is slightly unclear from the figure, and the river of smoke propagating between 5 and 6 September is not illustrated in the figure.
Also, please discuss about the uncertainty or biases between the MODIS derived AOD and that of the modeled.
Why is NORAD AOD higher than BBRAD AOD?

**Comment 10: Page 14 Figure 7** Figure caption: "Results are shown for the BBRAD and NORAD members starting at 00:00 UTC on 1 September 2017."
Please revise the caption for Figure 7 to clarify the timeline. Additionally, consider using a more distinct color, such as black, for the dust extinction contours, as the yellow is blending into the background plots.

**Comment 11: Page 16 Figure 9** Figure caption: "…the cyan contour the cloud fraction at 10 %..."

Do you mean 'at' or '>' 10%?

**Comment 12: Page 19 Line 338-341** "It is also consistent with the wind speed anomaly at 3.5 km amsl (Fig. 11d) with a positive anomaly ($> 3ms^{-1}$) superimposed on the positive AOD anomaly, and a negative wind speed anomaly ($< -3m\ s^{-1}$) where the negative AOD anomaly is located (Fig. 11c)."

The consistencies are not readily apparent, as the negative AOD anomaly depicted in Figure 11c does not correspond to the area with a negative wind speed anomaly at 3.5 km. Instead, the negative anomaly aligns with what is shown in Figure 11a. Could you please provide clarification?

**Comment 13: Page 19 Line 341** "Both Windpoort and Etosha are located in the area of negative wind speed anomaly."

Do you mean at an altitude of 3.5 km? Also, could you mark Etosha on the map for better reference? The rationale for using 3.5 km as the reference altitude is not entirely clear. Please clarify.

**Comment 14: Page 20 Figure 11c** Although there is a strong positive anomaly in dust emission over the region around Windpoort, the DOD does not exhibit any change. Please explain.

**Comment 15: Page 20 Line 352-355** "Further south, over southern Namibia and northwestern South Africa, a bipolar dust-AOD anomaly pattern (reversed with respect to the one further north) is seen, with positive anomalies (> 0.2) to the west and negative anomalies to the east. It is consistent with the dust emission anomaly pattern and likely due to the eastern shift of the river of smoke in the NORAD simulation and the associated change in low-level dynamics."

This is slightly unclear.

**Comment 16: Section 6.** Manuscript bounces about a bit from here onwards, referring back Fig. 5 and Fig. 3 frequently. Potentially some repetition of material in these sections which could be consolidated.

**Comment 17: Page 23 Line 396** "off BBA" or "of BBA"?

**Comment 18: Page 23 Line 397** Is it possible to avoid the word "twin" in "twin ensemble simulations."

**Comment 19: Page 23 Line 397-398** "one including the direct and semi-direct radiative effects of aerosols (BBRAD)"

This is only mentioned in the conclusion, please add the details in the methodology.

---

## Author Comment (AC1)

**Review of "The radiative impact of biomass burning aerosols on dust emissions over Namibia and the long-range transport of smoke observed during AEROCLO-sA" by Cyrille Flamant et al.**

egusphere-2023-2371

Dear Editor

Thanks for giving us the opportunity to further improve the quality of the paper. Thanks also to the reviewers for their help and benevolent comments. We have implemented all the necessary modifications requested by the reviewers.

Below, we provide a detailed point-by-point answer to the minor comments made by the two reviewers. Our answers to the comments appear in red, whereas additional text included in the revised MS appear in blue.

| Referee #1 |
| --- |

This study investigated the radiative impact of biomass burning aerosols on local and regional atmospheric circulation patterns using a mesoscale non-hydrostatic model. The authors first concluded that their Meso-NH simulations can reasonably reproduce key features of dynamics, thermodynamics and atmospheric composition fields as seen in in-situ and remote sensing observations during a dust emission episode on Sept. 5, 2017. By comparing simulations with and without BBA radiative effects, the authors found that omitting the radiative effects of BBA results in weaker AEJ-S, weaker LLJ, deeper convective PBL, and an eastward shift of the smoke plume.

I found the methodology and the scientific conclusions of this study sound and solid. This study is no doubt publishable with profound implications for air quality and dust cycle in the southern hemisphere. The manuscript is in general well written and easy to follow, however, I do find places where lengthy sentences demand several readthroughs to understand. I believe the readers would appreciate some extra work on these lengthy sentences.

I recommend publish after minor revision.

**General comments:**
**First,** I am impressed by the capability of Meso-NH in reproducing BBA spatiotemporal distributions and atmospheric circulation patterns during the 5-day time period. I have a few questions regarding the setup of the simulations:
1) is there any nudging or re-initialization of the simulation based on ERA5 during the course of the simulation? Is the model free-running for the whole time period since initialization?
There is no nudging or re-initialization of the simulation based on ECMWF operational analyses. The model runs freely for the entire period since initialization.
2) what is the rationale to create ensemble members by shifting initialization time? have you considered perturbing parameters related to BBA, e.g., SSA?
To clarify, it is now written "These two ensembles of five members allow us to isolate the significant changes in model response due to the BBA radiative effect from the internal model variability. In the following, the statistical significance of these changes is assessed using the two-tailed Student's t test (at the 95% confidence level)".
We did not consider perturbing parameters related to BBA. This could be considered in future studies.

3) what exactly is the vertical resolution? And where is the top of your domain (i.e., how do you define "*600m in the free troposphere*")?

Meso-NH is run over a domain covering both BBA sources over southern Africa and dust emissions over Namibia (Fig. 1c) using a horizontal resolution of 5 km and 64 levels in the vertical up to 25 km with a spacing of 60 m close to the surface, increasing to 600 m above 6 km.

This information is now added to the text.

4) how is "NORAD" achieved? by turning off SW calculations related to BBA in RRTM? How is the semi-direct effect (those related to cloud adjustments) excluded?

Yes, NORAD is achieved by turning off SW calculations related to BBA in the radiative code of Fouquart and Bonnel (1986).

The semi-direct effects of BBA are not excluded while the indirect effect is not considered here. We have added:

"It should be noted that the microphysical BBA–cloud interaction (the indirect effect of BBA) is not considered here."

I think these details could be added to the main text to help readers better understand the setup of the modeling framework.

Agreed. They are now included.

**Second,** I strongly recommend adding a schematic diagram to the manuscript, illustrating the complex pathways through which radiative effects of BBA modulate local and regional atmospheric circulation patterns and the distribution of smoke and dust emissions in southern Africa. When I read through section 6, I had to constantly flip through figures to build such a mechanistic diagram in my mind, I believe such a diagram in the main text would benefit the manuscript in profound ways.

We agree this is a nice way of synthetizing our findings. We have included the schematic below in the revised version of the MS **as Figure 15.**

[Figure]

[Figure]

Schematic representation of the impact of non-accounting for BBA direct and semi-direct effects over southern Africa on the low and mid-level atmospheric dynamics, namely: the strength of the LLJ, the development of the PBL, the strength of the AEJ-S, and the location of the river of smoke with respect to Etosha. Left: Main atmospheric dynamic components when the BBA radiative are taken into account. Right: same fields as left, but not taking into account the BBA radiative impact. The black arrow indicates the North. The dark grey arrow indicates the main orientation of the river of smoke. The red arrow indicates the vertical motion ahead/east of the river of smoke.

Other than these two points, I only have a few minor comments and suggestions that I encourage the authors to consider while revising the manuscript.

**Minor comments/suggestions:**
**Line 34-39**, additional reference for the importance of BBA direct and semi-direct effects: Diamond et al. (2022) pointed out the critical role of accounting for smoke diabatic heating that reduces the free-tropospheric subsidence in reproducing the observed low-cloud cover over the SE Atlantic with a regional model.
Thanks. We have added the reference to the work by Diamond et al. after the description of the findings of Chaboureau et al. (2022).
Diamond, M. S., Saide, P. E., Zuidema, P., Ackerman, A. S., Doherty, S. J., Fridlind, A. M., Gordon, H., Howes, C., Kazil, J., Yamaguchi, T., Zhang, J., Feingold, G., and Wood, R.: Cloud adjustments from large-scale smoke–circulation interactions strongly modulate the southeastern Atlantic stratocumulus-to-cumulus transition, Atmos. Chem. Phys., 22, 12113–12151, https://doi.org/10.5194/acp-22-12113-2022, 2022.

**Section 4.1**, Zhang & Zuidema (2021) also documented the important role of synoptic patterns in governing smoke transport over the oceanic region during the month of September, such that a mid-latitude intrusion pattern (their Fig. 8c, very similar to Fig. 2b in this manuscript) is often associated with less smoky conditions in the remote SE Atlantic, consistent with reduced transport of smoke by a weaker AEJ-S. They show that this synoptic pattern also strongly affects the extent of the marine stratocumulus deck.
This information was added to Section 4.1, after referring to Figure 2b, as:
"This synoptic pattern is often associated with less smoky conditions in the remote south eastern Atlantic (Zhang et Zuidema, 2021)."

Zhang, J. and Zuidema, P.: Sunlight-absorbing aerosol amplifies the seasonal cycle in low-cloud fraction over the southeast Atlantic, Atmos. Chem. Phys., 21, 11179–11199, https://doi.org/10.5194/acp-21-11179-2021, 2021.

**Fig. 2**, please indicate what "e" and "w" represent in the caption, please check the orientation of a)-d) in this figure.
W is for Windpoort (15.48°E, 19.37°S) and E is for Etosha (16.46°E, 18.77°S). This is now indicated in the Figure caption.

**Fig. 10**, there is no blue dotted line on the figure, you meant blue Xs, right?
That is correct, thanks for picking this up. This is now indicated in the Figure caption.

**Line 296-299**, this sentence is lengthy and hard to understand, I recommend rewording.
The sentence "Large extinction values are simulated below the LLJ with the largest near surface extinction values simulated amongst the location of the four dropsondes, while strong LNG-derived extinction values are observed just above the surface, these values also correspond to the largest ones observed at the dropsondes locations with the airborne lidar (Fig. 4p)." has been reworded as:
"Large extinction values are simulated below the LLJ, with the largest near-surface extinction values simulated amongst the location of the four dropsondes. Likewise, strong LNG-derived extinction values are observed just above the surface, which also correspond to the largest airborne lidar values detected at the locations of the dropsondes (Fig. 4p)."

**The section titles** in general seem a bit long to me, I think more concise and shorter section titles would read and look better.

We thought about it but could not convince ourselves to modify them: shortening did not help conveying the message we want to deliver.

**Line 311**, not sure if "*realistic*" is the right word here, I am convinced that the model captured the key features, but I feel "*realistic*" is perhaps too strong here.
Realistic is defined as "representing things in a way that is accurate and true to life." We believe this is in line with the message we would like to deliver regarding the "quality" of the Meso-NH simulations. We have kept 'realistic'.

**Line 330-331**, not clear which simulation set you're referring to. "*…is enhanced*" in which simulation? "*… associated low-level dynamics*" could you be more specific? Weaker or stronger LLJ?
The activation of dust sources is enhanced in NORAD (Fig. 7c) compared to BBRAD (Fig. 7b). This is related to the impact of the low-level dynamics associated with the river of smoke, not to be confused with the LLJ. The LLJ is weaker in the case of NORAD. This is now specified in the text.

**Line 355-358**, I do not see the temporal shift in Fig. 10f and h, to me, the orange and green curves pretty much track each other at Upington.
The shift may not be so evident in Figure 10h because of the time series is shown over 5 days and the shift is only by a few hours. But there is a shift on 5 September.

**Line 371-374**, not clear, please indicate "enhanced upward motion and drier mid-troposphere" in which simulation set? The cloud fraction increase is also not clear to me, I see both +ve and -ve cloud fraction changes, are you referring to the signal around 15 E or 25 E?
The vertical velocity signal is enhanced between 2 and 4 km amsl over Etosha pan (around 16-17°E, latitude of Etosha pan) in Fig. 12f in NORAD. The mid-troposphere is drier and the cloud fraction reduced (not increased) to the east of Etosha around 4 km amsl in NORAD.  This is now corrected.

**Line 375-376**, my interpretation of the above sentence and Fig. 12f is that upward motion in BBRAD is enhanced due to differential warming, but here, you indicate NORAD has enhanced vertical motion. I am confused. Also, could you be more specific about how the strength of updraft (vertical motion), LLJ and surface wind is dynamically related? (again, a schematic would help a lot).
Agreed. In fact, the upward motion is enhanced due to the eastward displacement of the river of smoke. The differential heating is just a manifestation of that. The eastward displacement also acts to weaken the LLJ over Etosha pan. We have added such a schematic as Figure 15 of the revised MS.

The sentence "The differential warming between BBRAD and NORAD in the BBA layer leads to enhanced upward vertical motion over Etosha (Fig. 12f) as well as a drier mid-troposphere (Fig. 12g) and increased cloud fraction (Fig. 12h) to the east consistent with the eastern displacement of the river of smoke in NORAD." has been modified in the revised MS as:
"The eastern displacement of the river of smoke leads to enhanced upward vertical motion over Etosha (around 16-17°E) between 2 and 4 km amsl (Fig. 12f) as well as to a drier mid-troposphere (Fig. 12g) and reduced cloud fraction (Fig. 12h) around 4 km amsl to the east of Etosha pan."
* * *
Referee #2

The authors of the study have conducted a comprehensive investigation into the radiative effects of biomass burning aerosols on atmospheric circulation at low and mid-levels over southern Africa. The study, contextualized within the AEROCLO-sA field campaign on September 5, 2017, utilized both in situ and remote sensing observations, alongside sets of ensemble simulations using the Meso-NH mesoscale model.

Their findings highlight that the simulation incorporating BBA radiative effects was shown to represent regional dynamics, thermodynamics, and compositional features more convincingly. In contrast, simulations neglecting these effects presented unrealistic dynamics, such as a weakly formed LLJ and inaccurately represented mid-level dynamics critical for the transport of BBAs. Specifically, the absence of BBA radiative effects resulted in discrepancies such as an inadequately established LLJ overnight and a convective planetary boundary layer (PBL) that was too deep in comparison with observational data. Additionally, the research connected the enhanced near-surface extinction coefficient values observed over Etosha with the downward mixing of BBAs in the convective boundary layer, rather than dust emission due to LLJ breakdown after sunrise.

The study concludes with a clear and important message that for accurate forecasting of dust emissions in Namibia, it is essential to consider the radiative effects of biomass burning aerosols, underscoring their significant influence on atmospheric behavior. However, I have some reservations about some aspects. It is good that this is a very detailed paper, but there is lot of bouncing backwards and forwards between figures in the results section. I would suggest removing some redundant details in the paper and streamlining the text. The details of the model configurations are somewhat missing. I recommend a minor revision before publication.

**Specific comments:**
**Comment 1: Page 3, line 59** "In particular, we detail the ensemble simulations designed with and without BBA radiative impact…"
Please provide a short overview of 'ensemble simulation' of what? In the introduction emphasize the importance of model simulations in the context of the current topic of interest. Include details that underscore the synergistic method combining both measurement and modeling techniques.
We now specify "Meso-NH ensemble simulations". Details are provided in the subsequent section 3.2. The synergetic methods used in this paper are classic and are based on detailed comparison of numerous atmospheric dynamics, thermodynamics and compositions fields.

**Comment 2:** Could you provide a very short background information on the overall data collection during the campaign and explain the reasons for selecting this particular day for the current analysis? This point seems somewhat obscured in the text and is not clearly articulated.
The complete of the data collected during the AEROCLO-sA field campaign is described in Formenti et al. (2019). It includes data collected from a ground-based supersite in Henties Bay as well as remote sensing and in situ observation acquired from the Safire Falcon 20. Here we focus on the dust emission episode over Etosha pan that occurred on 5 September. This is indicated in the text.

**Comment 3: Page 5, line 120** "Wiggs et al. (2022) have estimated the threshold wind velocity to be 7.25 ms−1 over Etosha during the dry season."
Can you explain what is meant by 'threshold wind velocity'?
This is defined at line 46 of the original MS, when the term is first introduced: "threshold wind velocity, i.e. the minimum wind speed initiating the wind erosion."

**Comment 4: Page 6, Section 3.2** Please add the details of the model configuration: boundary layer scheme, aerosol scheme, etc. Is the model's horizontal resolution set at 5 x 5 km, and does it reach a maximum altitude of 600 m? Please provide more details about BBRAD and NORAD to help the readers understand their relevance better.
The 1.5-order closure scheme for turbulence (Cuxart et al. 2000) and the eddy diffusivity mass-flux scheme for shallow convection (Pergaud et. 2009) are used to represent the boundary layer.
We changed "The horizontal spacing of 5 km" into "The horizontal resolution of 5 km" and added details on the vertical spacing of the grid on the vertical as "64 levels in the vertical up to 25 km with a spacing of 60 m close to the surface, increasing to 600 m above 6 km"
We added "The aerosol scheme is composed of two schemes, one for BBA and the other for dust."

Other details have been added as described in the reply to the first general comment of Referee#1.

**Comment 5: Page 7, line 169-171** "In early September, climatological mean…"
Could you specify the exact time period?
The period is 1997-2016 and was specified in the figure caption. It is now specified in the text.

**Comment 6: Page 8, Line 179-180** "over Angola and Zambia to the north are associated with the easterly flow along the northern fringes of the semi-permanent anticyclone (Fig. 2c)."
Is this the average at 07:00 UTC?
No, this is daily average for the climatological means. To clarify this, we now state that we are describing 'climatological' winds in that sentence. Also panels (c) and (d) of Figure 2 were inverted. This is now corrected in the revised version of the MS.

**Comment 7: Page 8, Figure 2.** Are 'E' and 'W' in the lower two panels referring to Etosha and Windpoort, respectively? Please mention this in the captions.
Yes, indeed. This information is now added to the caption.

**Comment 8: Page 9, Figure 3** Figure caption
Is this the result from one of the BBRAD ensemble members or the average of all the members? Are the values from the model hourly? If the model outputs are on an hourly basis, could the infrequent recording of the data introduce uncertainties when representing phenomena on a sub-hourly scale?
As written in the caption, "Results are shown for the BBRAD and NORAD members starting at 00:00 UTC on 1 September 2017." We added that "Results are shown every 30 min […]" so that the uncertainties introduced when representing phenomena on the sub-hourly scale are small.

**Comment 9: Page 13 Line 252-254** "The southeasterly BBA transport within the river of smoke is illustrated by the strong wind at 4 km amsl (Fig. 8b). The river of smoke propagated rapidly across southern Africa between 5 and 6 September 2017, under the influence of the fast evolving temperate tropical trough."
The first sentence is slightly unclear from the figure, and the river of smoke propagating between 5 and 6 September is not illustrated in the figure.
We modified the sentence as:
"The southeasterly BBA transport within the river of smoke is illustrated by the strong wind at 4 km amsl along the coast of Namibia (Fig. 8b). The river of smoke propagated rapidly across southern Africa between 5 and 6 September 2017, under the influence of the fast evolving temperate tropical trough as described in Flamant et al. (2022)."
Also, please discuss about the uncertainty or biases between the MODIS derived AOD and that of the modeled.
We feel this is beyond the scope of the study as we use MODIS data mainly to locate the position of the river of smoke.
Why is NORAD AOD higher than BBRAD AOD?
The weaker AEJ-S in the case of NORAD likely leads to more BBA accumulation at the border between Namibia and Angola.

**Comment 10: Page 14 Figure 7** Figure caption: "Results are shown for the BBRAD and NORAD members starting at 00:00 UTC on 1 September 2017."
Please revise the caption for Figure 7 to clarify the timeline. Additionally, consider using a more distinct color, such as black, for the dust extinction contours, as the yellow is blending into the background plots.
The first sentences of the caption read "Time-height cross-sections of extinction at 1064 nm on 5 September 2017 [...]. LNG observations were taken between 07:53 and 09:55 UTC, and Meso-NH

simulations are at 09:00 UTC […]". We believe this information gives a clear idea of the chronology. The sentence "Results are shown for the BBRAD and NORAD members starting at 00:00 UTC on 1 September 2017" explains which simulations are shown, in that case, those starting at 00:00 UTC. The dust extinction contours are now in black.

**Comment 11: Page 16 Figure 9** Figure caption: "…the cyan contour the cloud fraction at 10%..."
Do you mean 'at' or '>' 10%?
Yes, we mean at 10%.

**Comment 12: Page 19 Line 338-341** "It is also consistent with the wind speed anomaly at 3.5 km amsl (Fig. 11d) with a positive anomaly (> 3ms−1) superimposed on the positive AOD anomaly, and a negative wind speed anomaly (< −3m s−1) where the negative AOD anomaly is located (Fig. 11c)."
The consistencies are not readily apparent, as the negative AOD anomaly depicted in Figure 11c does not correspond to the area with a negative wind speed anomaly at 3.5 km. Instead, the negative anomaly aligns with what is shown in Figure 11a. Could you please provide clarification?
This is our mistake… Thanks for picking it up… AOD anomaly is shown in Figure 11a, not 11c. We have corrected 11c into 11a in the revised MS.

**Comment 13: Page 19 Line 341** "Both Windpoort and Etosha are located in the area of negative wind speed anomaly."
Do you mean at an altitude of 3.5 km?
Yes. This is now specified.
Also, could you mark Etosha on the map for better reference? The rationale for using 3.5 km as the reference altitude is not entirely clear. Please clarify.
This is a level where the anomalies are well structured. The wind anomalies at 4 km amsl are very similar.

**Comment 14: Page 20 Figure 11c** Although there is a strong positive anomaly in dust emission over the region around Windpoort, the DOD does not exhibit any change. Please explain.
Even though emission anomalies look significant thanks to the chosen scale [-0.5,0.5] g m$^{-2}$ d$^{-1}$, they are to week to actually show on the DOD anomaly map given the selected scale [-0.4,0.4].

**Comment 15: Page 20 Line 352-355** "Further south, over southern Namibia and northwestern South Africa, a bipolar dust-AOD anomaly pattern (reversed with respect to the one further north) is seen, with positive anomalies (> 0.2) to the west and negative anomalies to the east. It is consistent with the dust emission anomaly pattern and likely due to the eastern shift of the river of smoke in the NORAD simulation and the associated change in low-level dynamics."
This is slightly unclear.
We have removed "(reversed with respect to the one further north)" from the first sentence, for the sake of clarity.

**Comment 16: Section 6.** Manuscript bounces about a bit from here onwards, referring back Fig. 5 and Fig. 3 frequently. Potentially some repetition of material in these sections which could be consolidated.
We double-checked but could not find obvious repetitions in this section, which uses previous Figures (in particular 3 and 5) to consolidate the results based on Figures 11 -14.

**Comment 17: Page 23 Line 396** "off BBA" or "of BBA"?
Now corrected as "of BBA".

**Comment 18: Page 23 Line 397** Is it possible to avoid the word "twin" in "twin ensemble simulations."

Done, replaced twin, in the abstract and in the conclusion, by "two".

**Comment 19: Page 23 Line 397-398** "one including the direct and semi-direct radiative effects of aerosols (BBRAD)"
This is only mentioned in the conclusion, please add the details in the methodology.
These effects are mentioned in the abstract, but need to be mentioned in the main text also in Section 3.2. This is now done in the revised version of the MS. Details on the methodology have been added (see the reply to the Referee#1 first general comment).